# Opposing chemosensory functions of closely related gustatory receptors

Ji-Eun Ahn, Hubert Amrein*

Department of Cell Biology and Genetics, School of Medicine, Texas A&M University, Bryan, United States

**\*For correspondence:**
amrein@tamu.edu

**Competing interest:** The authors declare that no competing interests exist.

**Abstract** In the fruit fly *Drosophila melanogaster*, gustatory sensory neurons express taste receptors that are tuned to distinct groups of chemicals, thereby activating neural ensembles that elicit either feeding or avoidance behavior. Members of a family of ligand -gated receptor channels, the Gustatory receptors (Grs), play a central role in these behaviors. In general, closely related, evolutionarily conserved Gr proteins are co-expressed in the same type of taste neurons, tuned to chemically related compounds, and therefore triggering the same behavioral response. Here, we report that members of the Gr28 subfamily are expressed in largely non-overlapping sets of taste neurons in *Drosophila* larvae, detect chemicals of different valence, and trigger opposing feeding behaviors. We determined the intrinsic properties of *Gr28* neurons by expressing the mammalian Vanilloid Receptor 1 (VR1), which is activated by capsaicin, a chemical to which wild-type *Drosophila* larvae do not respond. When VR1 is expressed in *Gr28a* neurons, larvae become attracted to capsaicin, consistent with reports showing that *Gr28a* itself encodes a receptor for nutritious RNA. In contrast, expression of VR1 in two pairs of *Gr28b.c* neurons triggers avoidance to capsaicin. Moreover, neuronal inactivation experiments show that the *Gr28b.c* neurons are necessary for avoidance of several bitter compounds. Lastly, behavioral experiments of *Gr28* deficient larvae and live Ca$^{2+}$ imaging studies of *Gr28b.c* neurons revealed that denatonium benzoate, a synthetic bitter compound that shares structural similarities with natural bitter chemicals, is a ligand for a receptor complex containing a Gr28b.c or Gr28b.a subunit. Thus, the *Gr28* proteins, which have been evolutionarily conserved over 260 million years in insects, represent the first taste receptor subfamily in which specific members mediate behavior with opposite valence.

## eLife assessment

This **valuable** study focuses on the role of the Gr28 family of insect chemoreceptors. Using the *Drosophila* larva, the authors show that taste neurons expressing different members of this family of bitter taste receptors trigger opposite behavior – attraction and repulsion. They establish the minimal bitter taste receptor subunit composition needed in these neurons to mediate the repulsion of bitter tastants. The evidence presented is **convincing**, using well-validated and controlled tools and experiments.

## Introduction

Meaningful animal behavior is established through cooperative engagement of multiple sensory systems. In many insects, the chemosensory system plays a central role in such integration processes. The fruit fly *Drosophila melanogaster* has served as the primary insect model system for elucidating the molecular basis and neural circuitry of both olfaction and taste, by virtue of the vast genetic resources, amenability to both neurophysiological recording and live imaging, and simple yet powerful

behavioral assays, allowing investigators to link genes to chemosensory behavior and neural activity (*Montell, 2021*).

The *Drosophila* gustatory system is characterized by several insect-specific anatomical and molecular features. For example, taste cells are primary sensory neurons, with dendritic processes that express taste receptors, while long axons project and convey taste information directly to the brain. In adult flies, these neurons, referred to as Gustatory Receptor Neurons (GRNs), are distributed across several appendages, such as labial palps, legs, and presumably the antennae. Additionally, some GRNs reside internally, arranged in cell clusters along the pharynx (*Amrein, 2016*; *Joseph and Carlson, 2015*; *Scott, 2018*). Likewise, *Drosophila* larvae have numerous structures located both on the head surface and internally along the larval pharynx, where chemical compounds are assessed during their passage toward the digestive system (*Figure 1A*; *Apostolopoulou et al., 2015*; *Kwon et al., 2011*; *Rist and Thum, 2017*).

Fruit flies, like most insects, employ taste receptors encoded by two major gene families, the *Gustatory receptor* (*Gr*) and the *Ionotropic Receptor* (*IR*) genes, to sense soluble chemicals, such as appetitive food compounds, noxious and toxic chemicals, as well as pheromones. Both Gr- and IR-based receptors are thought to form complexes composed of several, and generally different, subunits. Composition of only a few Gr-based taste receptors for sugars and bitter compounds is known (*Jiao et al., 2008*; *Yavuz et al., 2014*; *Shim et al., 2015*; *Sung et al., 2017*; Fujii et al., unpublished), and two Gr proteins, Gr21a and Gr63a, are co-expressed in a small subset of olfactory neurons where they form a receptor complex for gaseous carbon dioxide (*Jones et al., 2007*; *Kwon et al., 2007*).

The *Gr* genes represent the largest taste receptor gene family in insects. In *D. melanogaster*, it is comprised of 60 genes predicted to encode 68 proteins, expression of which has been extensively described (*Dahanukar et al., 2007*; *Dunipace et al., 2001*; *Fujii et al., 2015*; *Scott et al., 2001*; *Weiss et al., 2011*). Several *Gr* genes have been functionally characterized in adult flies using genetic mutations combined with either electrophysiological recordings, $Ca^{2+}$ imaging studies, or behavioral analyses (*Amrein, 2016*; *Montell, 2021*), but only a few have been studied in larvae (*Apostolopoulou et al., 2016*; *Choi et al., 2016*; *Choi et al., 2020*; *Mishra et al., 2013*; *Mishra et al., 2018*).

The *Gr28* gene subfamily is of particular interest for a number of reasons: first, it is one of the most conserved *Gr* subfamilies, homologs of which can be found across all insect families and even more distant arthropods (*Eyun et al., 2017*; *Fujii et al., 2023*; *Suzuki et al., 2018*). The six *Gr28* genes are tightly clustered, with the five *Gr28b* genes transcribed from distinct promotors and unique exons spliced to shared second and third exons, while *Gr28a* is a separate transcription unit (*Figure 1B*). Overall conservation between the Gr28 proteins is high (≥50% similarity), characteristic for genes generated through recent gene duplication events. Second, the *Gr28* genes are expressed not only in the gustatory system of larvae and adult flies, but also in many other organs, especially the central nervous system and non-chemosensory neurons of the peripheral nervous system, suggesting that they have functions beyond gustation and are important to sense chemical signals unrelated to food (*Mishra et al., 2018*; *Thorne and Amrein, 2008*). Notably, evidence for such roles has been reported before any direct link to gustatory perception was discovered. Ni and colleagues showed that *Gr28b.d* is essential for high-temperature avoidance in flies (*Ni et al., 2013*), while Xiang and collaborators found that *Gr28* mutant larvae were deficient in UV light avoidance (*Xiang et al., 2010*). Third, the only known gustatory function for any Gr28 protein thus far is sensing of RNA and ribose by Gr28a, mediated by *Gr28a-GAL4* GRNs (*Mishra et al., 2018*). Remarkably, sensing RNA and ribose is an appetitive taste quality found in other dipteran insects that diverged from *Drosophila* about 260 million years ago, including flesh flies and mosquitoes, and we showed that *Gr28* homologs from *Aedes aegypti* and *Anopheles gambiae* can restore RNA and ribose preference in *Drosophila Gr28* mutant larvae when expressed in *Gr28a-GAL4* GRNs (*Fujii et al., 2023*).

Here, we present a detailed expression analysis and functional characterization of the *Gr28* genes and the respective GRNs in *Drosophila* larvae. In addition to *Gr28a*, three of the five *Gr28b* genes (*Gr28b.a*, *Gr28b.c*, and *Gr28b.e*) are also expressed in the larval taste system. Interestingly, GRNs expressing *Gr28a* and the three *Gr28b* genes represent functionally distinct neuronal ensembles, with minimal expression overlap in a single pair of neurons expressing both *Gr28a* and *Gr28b.c*. When the mammalian Vanilloid Receptor 1 (VR1) is expressed under the control of specific GAL4 drivers, we found that *Gr28a-GAL4* and *Gr28b.c-GAL4* neurons mediate opposing taste behavior in the presence of capsaicin, the ligand for VR1. Specifically, *Gr28a-GAL4/UAS-VR1* larvae show strong attraction

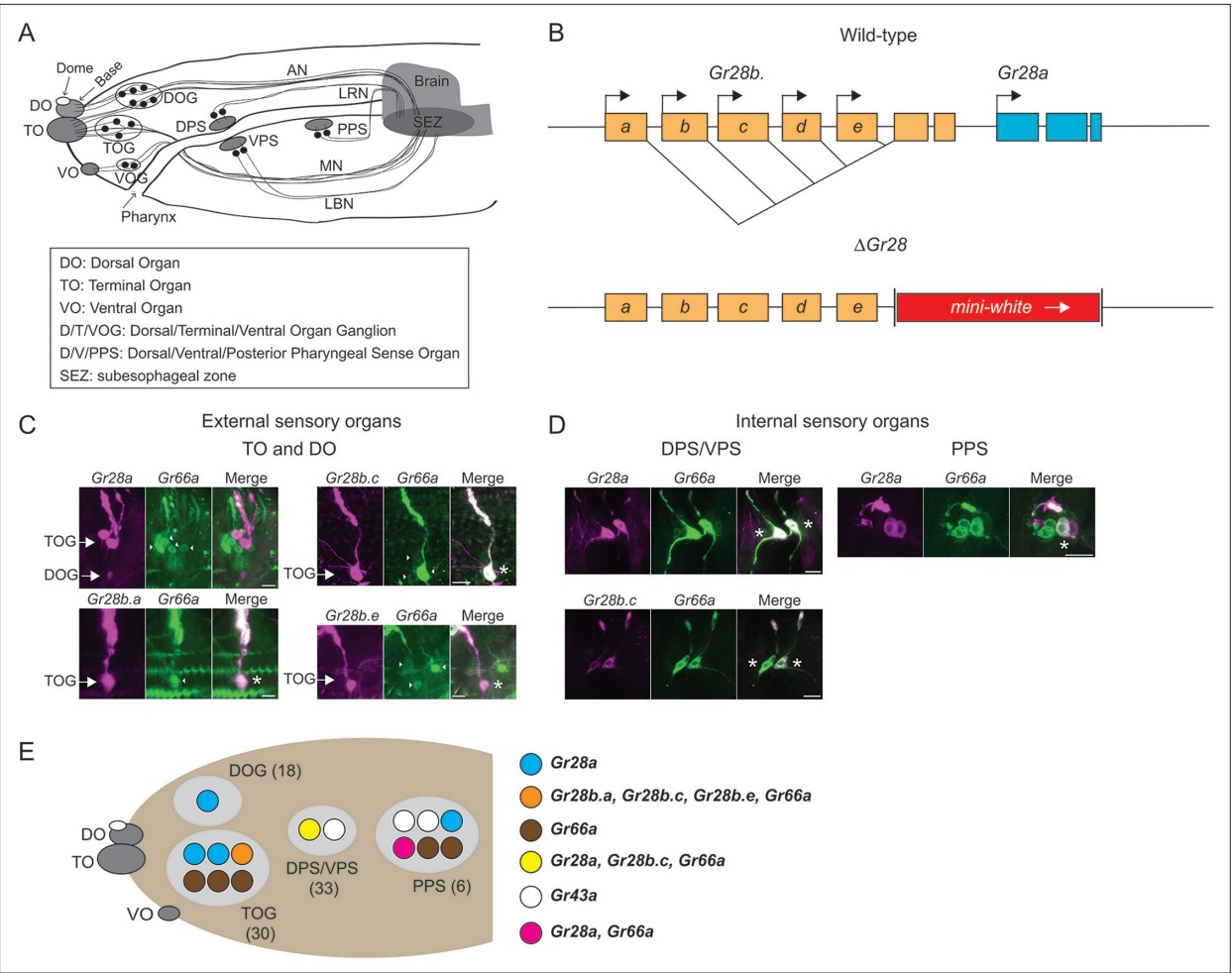

**Figure 1.** Expression of the *Gr28* genes in the larval sensory organs. (**A**) Schematic representation of the larval chemosensory system. Three external sensory organs (dorsal organ [DO], terminal organ [TO], and ventral organ [VO]) hold collectively the dendritic extensions of neuronal cell bodies in the respective ganglia (dorsal organ ganglia [DOG], terminal organ ganglia [TOG], and ventral organ ganglia [VOG]). Three clusters of sensory neurons (dorsal pharyngeal sense organ [DPS], ventral pharyngeal sense organ [VPS], and posterior pharyngeal sense organ [PPS]) reside along the pharynx. The antennal nerve (AN) connects the DOG neurons to the subesophageal zone (SEZ). The TOG and VOG neurons project along the maxillary nerve (MN) to the SEZ. The DPS and PPS neurons project axons to the SEZ via the labral nerve (LRN). The VPS neurons project to the SEZ through the labial nerve (LBN). Note that olfactory neurons are omitted in the schematic here and in (**E**) (below). (**B**) Structure of the *Gr28* locus. The six *Gr28* genes are clustered within 10 kilobases. The five *Gr28b* genes are transcribed from distinct promotors, with first unique exons that are spliced to common second and third exons. *Gr28a* is downstream of the *Gr28b* genes. The *Gr28* mutation (*ΔGr28*) used in this study lacks the shared common exons of the *Gr28b* genes and the entire *Gr28a* gene. (**C**) Expression of the *Gr28* genes in the external sensory organs. Note that images show only one of the bilaterally symmetrical organs. Co-expression between *Gr66a* (a marker for bitter taste gustatory receptor neurons [GRNs]) and different *Gr28* genes (in 'Merge' panel) was assessed using *GAL4* and *LexA* drivers for *Gr28* genes and *Gr66a*, respectively. For *Gr28b.a*, *Gr28b.c* and *Gr28b.e* co-expression was observed in each case. However, *Gr66a* and *Gr28a* are expressed in different GRNs. Asterisks refer to a GRN expressing both *Gr66a* and the indicated *Gr28* gene. Scale bars are 5 µm. (**D**) Expression of the *Gr28* genes in internal sensory organs. Note that the bilaterally symmetrical halves of the DPS/VPS are physically close to each other, and the images includes both halves, while the image of the PPS shows only one side of the bilaterally symmetrical organ. *Gr28b.c*, but none of the other *Gr28b* genes, is co-expressed with *Gr66a* in the DPS/VPS. In the PPS, none of the *Gr28b* genes is found, while *Gr28a* and *Gr66a* are partially co-expressed, but each gene is also expressed exclusively in a subset of GRNs. Asterisks refer to a GRN expressing both *Gr66a* and the indicated *Gr28* gene. Scale bars are 5 µm. (**E**) Expression summary: only relevant neurons in one of the paired taste organs are shown, with total number of neurons indicated in parenthesis. The cartoon summarizes the immunostainings shown in (**C**) and (**D**). The VOG is not shown as none of the *Gr28-GAL4* drivers is expressed there. Immunostaining was performed on whole-mount preparations from larvae heads of the following genotypes: *UAS-mCD8:RFP lexAop-rCD2:GFP;Gr66a-LexA/Gr28a-GAL4*, *UAS-mCD8:RFP lexAop-rCD2:GFP;Gr66a-LexA/Gr28b.c-GAL4*, *UAS-mCD8:RFP lexAop-rCD2:GFP;Gr66a-LexA/Gr28b.e-GAL4* and *UAS-mCD8:RFP lexAop-rCD2:GFP;Gr66a-LexA/+; Gr28b.a-GAL4/+*. The *Gr43a^GAL4* GRNs, which do not overlap with *Gr28a-GAL4* neurons (***Mishra et al., 2018***), are shown for reference to experiments described in (***Figure 2***).

The online version of this article includes the following figure supplement(s) for figure 1:

**Figure supplement 1.** Co-expression analysis between *Gr28* genes in larval sensory organs.

for capsaicin, while *Gr28b.c-GAL4/UAS-VR1* larvae show strong avoidance of capsaicin. Neuronal inactivation experiments reveal that the *Gr28b.c* GRNs are necessary to sense bitter compounds, such as denatonium, quinine, lobeline, and caffeine. Moreover, $Ca^{2+}$ responses of *Gr28b.c* GRNs to denatonium and quinine are significantly reduced and avoidance behavior of these two chemicals is diminished in *Gr28* mutant larvae, whereas $Ca^{2+}$ responses and avoidance behavior were not affected when challenged with lobeline and caffeine. This implies that Gr28b proteins are subunits of receptor complexes sensing a subgroup of bitter tasting compounds. In summary, the *Gr28* genes encode related Gr proteins mediating both positive and negative valence.

## Results
### Expression of the *Gr28* genes in the larval taste organs

The peripheral chemosensory system of the larvae is subdivided into bilaterally symmetrical, 'external' and 'internal' taste organs (*Stocker, 2008*). The three external organs reside near the tip of the head and are organized as paired ganglia, the dorsal, terminal, and ventral organ ganglia (DOG, TOG, and VOG) that house the GRN cell bodies with dendritic extensions in respective sensory organs (DO, TO, and VO) at the head surface, while carrying information via their axons to the subesophageal zone (SEZ) in the brain (*Figure 1A*). The DOG harbors 21 olfactory neurons and 18 presumptive GRNs. For clarity, GRN numbers refer to neurons in one of the two, bilaterally symmetrical taste organs. The GRNs located in the DOG fall into two distinct groups, based on their dendritic extensions: 11 presumptive GRNs, 4 of which were shown to express *Gr* genes (*Apostolopoulou et al., 2015*), send dendrites to the base of the DO, while 7 neurons extend dendrites to the TO (the dorsolateral group, *Figure 1A*; *Kwon et al., 2011*). The TOG contains 30 neurons, with dendrites located in the TO (the distal group). The internal taste structures, referred to as the dorsal/ventral pharyngeal sense organ (DPS/VPS, 33 neurons) and the posterior pharyngeal sense organs (PPS, 6 neurons) are located along the pharynx and sense chemicals as they are moved toward the digestive system. We note that not all these neurons are confirmed GRNs, either by function or expression of *Gr* or *IR* genes, albeit based on their location and anatomy, most are thought to be GRNs (*Rist and Thum, 2017*; *Sánchez-Alcañiz et al., 2018*; *Stewart et al., 2015*).

Our initial expression analysis of the *Gr28* genes revealed that four of the six *Gr28* genes (*Gr28a*, *Gr28b.a*, *Gr28b.c,* and *Gr28b.e*) were expressed in larval taste organs, in addition to cells in the gut, the brain, and non-chemosensory cells of the larvae (*Mishra et al., 2018*; *Thorne and Amrein, 2008*). This and all previous *Gr* expression studies were performed using bimodal expression systems, mostly *GAL4/UAS*, whereby *Gr* promotors driving *GAL4* are assumed to faithfully reproduce expression of the respective *Gr* genes. Importantly, we analyzed two to four *Gr28-GAL4* insertion lines for each transgene, and at least two generated the same expression pattern (*Mishra et al., 2018*; *Thorne and Amrein, 2008*), providing evidence that the drivers reflect a fairly accurate expression profile of respective endogenous genes. To further delineate the putative chemosensory roles of these genes, we performed a more detailed co-expression analysis between the *Gr28* genes and the bitter taste receptor gene *Gr66a* by combining the *GAL4/UAS* (labeling *Gr28* expressing neurons) with the *LexA/lexAop* (marking *Gr66a* neurons) system (*Figure 1C and D*). In the TOG, we found expression of all four *Gr28* genes, along with that of *Gr66a*, which was expressed in three or four neurons (this number is slightly smaller than the six neurons previously reported by *Kwon et al., 2011*). *GAL4* drivers for *Gr28b.a*, *Gr28b.c*, and *Gr28b.e* are co-expressed in a single neuron with *Gr66a-LexA* (*Figure 1C*). In contrast, *Gr28a-GAL4* is expressed in a distinct TOG neuron than *Gr66a-LexA* (*Figure 1C*) and the *Gr28b* genes, an observation we independently confirmed using a *Gr28b.c-LexA* driver (*Figure 1—figure supplement 1*). In the DOG, we find only a single *Gr28a-GAL4* neuron, while none of the *Gr28b* genes, or *Gr66a*, is expressed there (*Figure 1C*). None of the *Gr28-GAL4* drivers was expressed in the VOG. In the internal sensory organs, *Gr66a-LexA* is found in one GRN in the DPS/VPS, where it is co-expressed with *Gr28b.c-GAL4* as well as *Gr28a-GAL4* (*Figure 1D*). In the PPS, none of the *Gr28b* genes is expressed, but *Gr28a-GAL4* is found in two neurons, one of which also expresses *Gr66a-LexA* (*Figure 1D*). In summary, the internal sensory organs can be subdivided into three distinct groups (*Figure 1E*): *Gr66a/Gr28b.c/Gr28a*-positive neurons, *Gr66a/Gr28a*-positive neurons, and *Gr28a*^only neurons, which contrasts the external sensory organs where *Gr28a* and the *Gr28b* genes are expressed in a mutually exclusive fashion.

## Subsets of *Gr28* neurons mediate opposing feeding behaviors

We previously showed that at least one of the six *Gr28* genes is necessary for feeding attraction to RNA, ribonucleosides, and ribose using a well-established two-choice feeding preference assay (*Mishra et al., 2018*; *Figure 2A*). Specifically, larvae homozygous mutant for *Gr28* (*ΔGr28*, a deletion of the entire *Gr28a* gene and more than half of the coding region of all *Gr28b* genes, *Figure 1B*) lose their ability to sense these compounds, a phenotype that is restored when single *UAS-Gr28* reporter transgenes are expressed in *Gr28a-GAL4* neurons. The largely non-overlapping expression of *Gr28a* and the *Gr28b* genes suggests that respective neurons represent functionally distinct entities. To investigate this possibility, we took advantage of the mammalian VR1 protein, a TRP channel that is activated by capsaicin (*Caterina et al., 1997*). *Drosophila* have no *VR1* like-gene in their genome and do not respond to capsaicin behaviorally, but flies are attracted to this chemical when a modified *VR1* gene (*VR1E600K*) is expressed in sweet taste neurons (*Marella et al., 2006*). Thus, we expressed *VR1E600K* (henceforth referred to *UAS-VR1*) under the control of the four *Gr28-GAL4* drivers (*Gr28a-GAL4*, *Gr28b.a-GAL4*, *Gr28b.c-GAL4,* and *Gr28b.e-GAL4*) in larvae and tested their response to capsaicin using the two-choice preference assay (*Figure 2A*). Additionally, we expressed *UAS-VR1* in bitter neurons as well as appetitive fructose sensing neurons using respective *GAL4* drivers (*Gr66a-GAL4* and *Gr43a^{GAL4}*) (*Mishra et al., 2013*; *Scott et al., 2001*), which served as control larvae (note that fructose sensing *Gr43a^{GAL4}* neurons are distinct from *Gr28a-GAL4* neurons; *Mishra et al., 2018*). Just like adult flies, *w^{1118}* control larvae lacking either a *UAS-VR1* reporter, a *Gr-GAL4* driver, or both, were unresponsive to 0.1 mM capsaicin, while the positive control larvae expressing *UAS-VR1* in either fructose sensing GRNs or bitter taste GRNs showed robust attraction to or avoidance of capsaicin (*Figure 2B*). Consistent with previous findings, which identified *Gr28a* GRNs as appetitive neurons (*Mishra et al., 2018*), larvae expressing *UAS-VR1* under the control of *Gr28a-GAL4* showed strong appetitive responses to capsaicin (*Figure 2B*). In contrast, larvae expressing *UAS-VR1* under the control of *Gr28b.c-GAL4* showed robust avoidance behavior to capsaicin, while expression in the single TOG GRN using either *Gr28b.a-GAL4* or *Gr28b.e-GAL4* (also expressing *Gr28b.c*; *Figure 1C*) caused neither attraction to nor avoidance of capsaicin.

Overlap between *Gr28a* and *Gr66a* in internal GRNs raises the question about their contribution to appetitive and/or avoidance behavior. We deemed it unlikely that these neurons were critical for appetitive behavior since the one pair located in the DPS/VPS (expressing also *Gr28b.c*) is necessary for capsaicin avoidance (see above). Indeed, when VR1 was suppressed in all *Gr66a/Gr28b.c* neurons by means of the GAL4 suppressor GAL80, these larvae, expressing VR1 in *Gr28a^{only}* GRNs, remained strongly attracted to capsaicin (*Figure 2B*, last panel; for effective suppression, see *Figure 2—figure supplement 1*). Taken together, this analysis indicates that *Gr28a-GAL4* is expressed in appetitive-inducing neurons, while *Gr28b.c-GAL4* neurons mediate avoidance behavior.

The capsaicin experiments above suggest that the two larval *Gr28b.c-GAL4* GRNs mediate negative valence. Previous reports have shown that larvae avoid many bitter compounds (*Apostolopoulou et al., 2016*; *Apostolopoulou et al., 2015*; *Choi et al., 2020*; *Choi et al., 2016*; *van Giesen et al., 2016*), and thus, we expected that eliminating activity of *Gr28b.c-GAL4* GRNs would result in loss of avoidance behavior to at least some bitter chemicals. Indeed, when *Gr28b.c-GAL4* GRN activity was blocked using the inward-rectifying potassium channel Kir2.1 (*Baines et al., 2001*; *Paradis et al., 2001*), larvae no longer avoided the four tested bitter compounds denatonium, quinine, lobeline, and caffeine (*Figure 3*). A compound-specific avoidance phenotype to lobeline and caffeine was observed when the *Gr28b.e-GAL4* GRNs in the TOG were inactivated. These data, together with the capsaicin experiments, suggest that two pairs of GRNs, one in the TOG and one in the DPS/VPS, are necessary and sufficient for avoidance of these four bitter tasting chemicals.

## Gr28b.c and Gr28b.a are subunits of a taste receptor complex for denatonium

We next examined whether any of the Gr28b proteins is part of a taste receptor complex detecting any of these bitter chemicals (*Figure 4*). Surprisingly, only avoidance of denatonium was affected in larvae lacking the *Gr28* gene cluster (*ΔGr28*; *Figure 1B*), while avoidance to quinine was somewhat reduced, albeit not significantly, and avoidance to lobeline or caffeine remained robust. In fact, avoidance to caffeine increased modestly, but significantly (*Figure 4A*). We then examined whether any of the Gr28 proteins was sufficient to restore denatonium avoidance by expressing individual *Gr28* genes

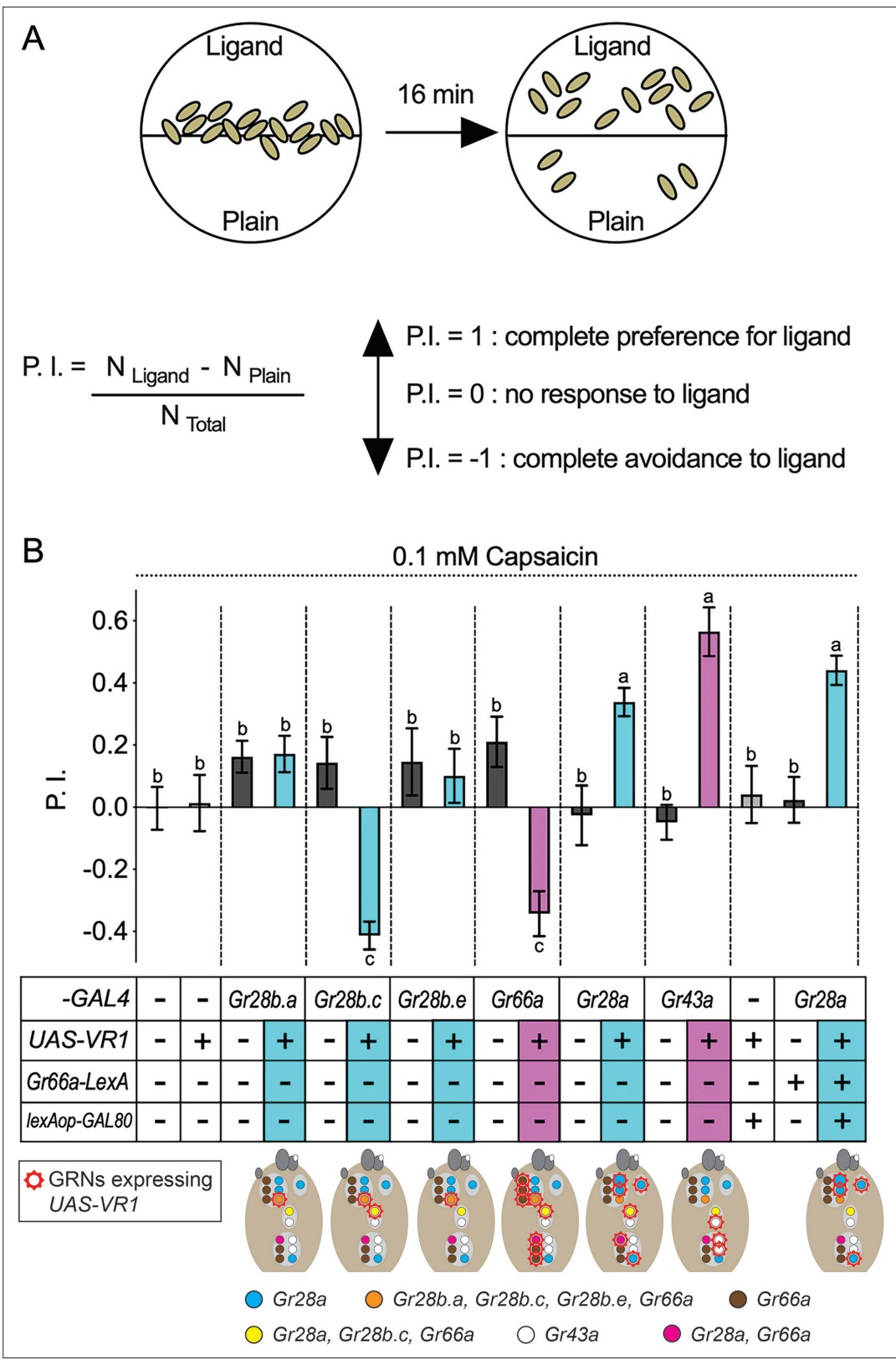

**Figure 2.** Intrinsic valence of different *Gr28* gustatory receptor neurons (GRNs). (**A**) Behavioral two-choice preference assay to quantify appetitive and avoidance of larvae for chemical ligands. Fifteen feeding stage, third-instar larvae are placed along the demarcation of a 1% agarose containing dish (35 mm), with one side plain, and the other side containing ligand. The preference index (P.I.; see 'Materials and methods') is calculated

*Figure 2 continued*

after counting location of larvae after 16 min. (**B**) *Gr28a* GRNs mediate capsaicin preference, while *Gr28b.c* GRNs elicit capsaicin avoidance in larvae. GRNs expressing the capsaicin receptor VR1 are marked with a red crown in diagrams below. $w^{1118}$, reporter gene only (*UAS-VR1/+*), and respective *GAL4* driver only (*Gr-GAL4/+*) larvae serve as negative controls (white panels) and show neither preference for nor avoidance to 0.1 mM capsaicin. Experimental larvae expressing VR1 in *Gr28-GAL4* neurons are shown in blue. Positive control larvae (purple panels) expressing VR1 in bitter taste GRNs (*Gr66a-Gal4*) or appetitive, sweet GRNs (*Gr43a^GAL4^*) show expected avoidance to or preference for capsaicin. Experimental larvae (blue panels) expressing VR1 in *Gr28a-GAL4* GRNs display robust preference for capsaicin, which is still observed when expression is further restricted to *Gr28a^only^* GRNs (*Gr66a-LexA/UAS-VR1; Gr28a-GAL4/lexAop-GAL80*; right panel). In contrast, when VR1 is expressed in *Gr28b.c-GAL4* GRNs, larvae strongly avoid capsaicin. Neither avoidance nor preference was observed when VR1 is expressed in the *Gr28b.a-GAL4* or *Gr28b.e-GAL4* GRNs. Each bar represents the mean ± SEM of P.I. (n = 10–22 assays). The taste behavior of *Gr-GAL4>UAS-VR1* larvae is compared to three controls ($w^{1118}$, *UAS-VR1/+* and *Gr-GAL4/+*) using one-way ANOVA with Bonferroni's multiple comparison tests (p<0.05), whereby different letters indicate a statistically significant difference. Dashed lines delineate groups for ANOVA. All control and experimental larvae are in the $w^{1118}$ background, carry one copy of the indicated transgene(s), and were generated from crosses of respective strains listed in 'Materials and methods'.

The online version of this article includes the following source data and figure supplement(s) for figure 2:

**Source data 1.** Taste preference assay for 0.1 mM capsaicin of larvae expressing VR1 in different GRNs.

**Figure supplement 1.** Suppression of GAL4 in a subset of *Gr28a-GAL4* neurons.

**Figure supplement 1—source data 1.** Expression of *GCaMP6m* limited to a subset of *Gr28a* GRNs using *lexAop-GAL80* under control of *Gr66a-LexA*.

**Figure supplement 1—source data 2.** Quantification of GFP positive GRNs recorded from images of *Gr28a* GRNs expressing *UAS-GCaMP6m* (A).

under the control of the *Gr28b.c-GAL4* driver. Indeed, either *Gr28b.a* or *Gr28b.c* expression led to a full recovery of denatonium avoidance, while expression of any other *Gr28b* gene, or *Gr28a*, failed to do so (*Figure 4B*). This observation suggests that despite the high level of similarity between these receptors, recognition of denatonium is dependent on specific structural features present in Gr28b.a and Gr28b.c, but not in any of the other Gr28 proteins.

Since *Gr66a-LexA* is co-expressed in all *Gr28b-GAL4*-expressing GRNs, we wondered whether Gr66a is a component of the denatonium receptor. Previous work had established that *Gr66a* is required for caffeine avoidance in both larvae and adult flies (*Apostolopoulou et al., 2016*; *Lee et al., 2009*; *Moon et al., 2006*), which we confirmed (*Figure 4—figure supplement 1*). Surprisingly, avoidance of denatonium and quinine was not diminished, but increased significantly (*Figure 4—figure supplement 1*). Given the multimeric nature of bitter taste receptors (*Sung et al., 2017*), one possibility is that the absence of a Gr subunit not required for the detection of denatonium (Gr66a) could favor formation of multimeric complexes containing Gr subunits that recognize this compound (such as Gr28b.a and/or Gr28b.c).

Finally, we investigated neuronal responses in larvae expressing the $Ca^{2+}$ indicator GCaMP6m in *Gr28b.c-GAL4* GRNs (*Figure 5*). We developed a whole animal imaging preparation, whereby larvae were placed in an 'imaging chamber' to minimize head movements (*Figure 5A*), and visualized neural activity of the *Gr28b.c-GAL4* GRN in the TOG in real time (*Chen et al., 2013*) upon exposure to the four bitter compounds, as well as sucrose, ribose, and fructose (*Figure 5B–D*). All bitter compounds elicited rapid $Ca^{2+}$ increases in *Gr28b.c-GAL4* GRNs, while none of the sugars did (*Figure 5C and D*). When neural activity was recorded in *Gr28b.c-GAL4* GRNs of *ΔGr28* homozygous mutant larvae (*Figure 5E*), $Ca^{2+}$ responses to denatonium and quinine were severely reduced, while responses to both caffeine and lobeline were not affected. Re-expression of either *Gr28b.a* or *Gr28b.c*, but not *Gr28b.b*, *Gr28b.d*, *Gr28b.e*, or *Gr28a* rescued $Ca^{2+}$ response to denatonium, but not to quinine (*Figure 5F and G*). Together, these experiments identified *Gr28b.c* and *Gr28b.a* as redundant subunits

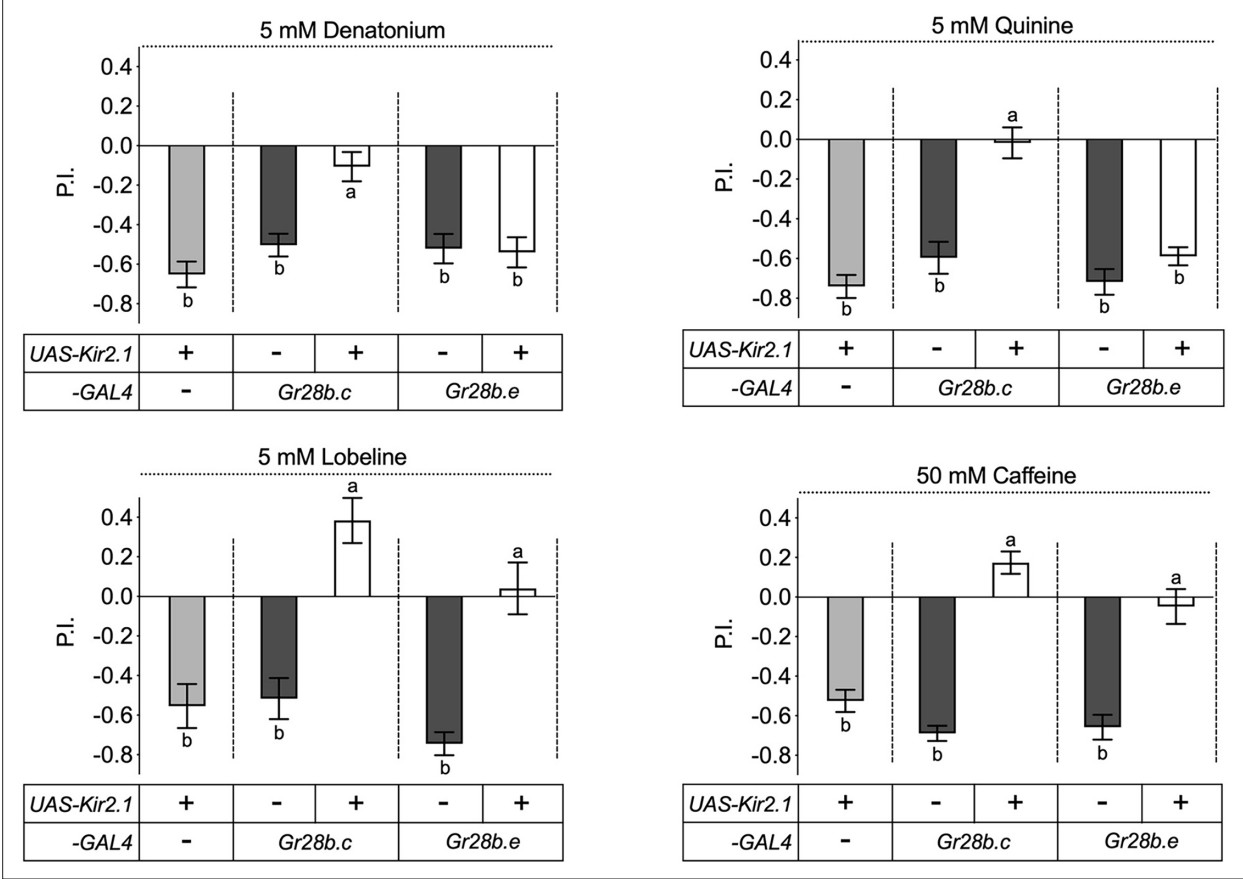

**Figure 3.** The *Gr28b.c* neurons mediate avoidance behavior to bitter compounds. Inactivation of *Gr28b.c* gustatory receptor neurons (GRNs) (*Gr28b.c-GAL4/UAS-Kir2.1*) elicits significantly reduced avoidance of larvae to all four bitter compounds tested – denatonium, quinine, lobeline, and caffeine – while control larvae, carrying either the driver or the reporter only, showed strong avoidance of these compounds. In contrast, larvae with inactivated *Gr28b.e* GRNs (*Gr28b.e-GAL4/UAS-Kir2.1*) still avoid denatonium and quinine (top), but no longer avoid lobeline and caffeine (bottom). Each bar represents the mean ± SEM of preference index (P.I.) (n = 11–20 assays). The taste behavior of *Gr28b.c-GAL4; UAS-Kir2.1* and *Gr28b.e-GAL4; UAS-Kir2.1* larvae was compared to two controls (*UAS-Kir2.1/+* and *Gr28b.c -GAL4/+* or *Gr28b.e-GAL4/+*) using Kruskal–Wallis test by ranks with Dunn's multiple comparison tests (p<0.05). Bars with different letters are significantly different. Dashed lines delineate groups for ANOVA. Fly genotypes: *w[1118]; UAS-Kir2.1/+* (light gray), *w[1118]; Gr28b.c-GAL4/+* (dark gray), *w[1118]; Gr28b.c-GAL4/UAS-Kir2.1* (white), *w[1118]; Gr28b.e-GAL4/+* (dark gray), *w[1118]; Gr28b.e-GAL4/UAS-Kir2.1* (white).

The online version of this article includes the following source data for figure 3:

**Source data 1.** Taste preference assay for bitter compounds of larvae with inactivated *Gr28b.c* or *Gr28b.e* GRNs using expression of *UAS-Kir2.1*.

of a denatonium receptor complex, a complex that does not require *Gr66a* or any of the other *Gr28b* subunits.

## Discussion

The Gr28 receptors comprise six related Gr proteins (*Figure 6*), forming one of the few Gr subfamilies conserved across diverse insect species (*Agnihotri et al., 2016*; *Engsontia and Satasook, 2021*; *Yu et al., 2023*). Yet, they were the least characterized when compared to other conserved subfamilies, such as the sugar receptors (Gr5a, Gr61a, and Gr64a-f), the carbon dioxide receptors Gr21a and Gr63a, or the bitter taste receptors. The only ligands associated with the Gr28 proteins were

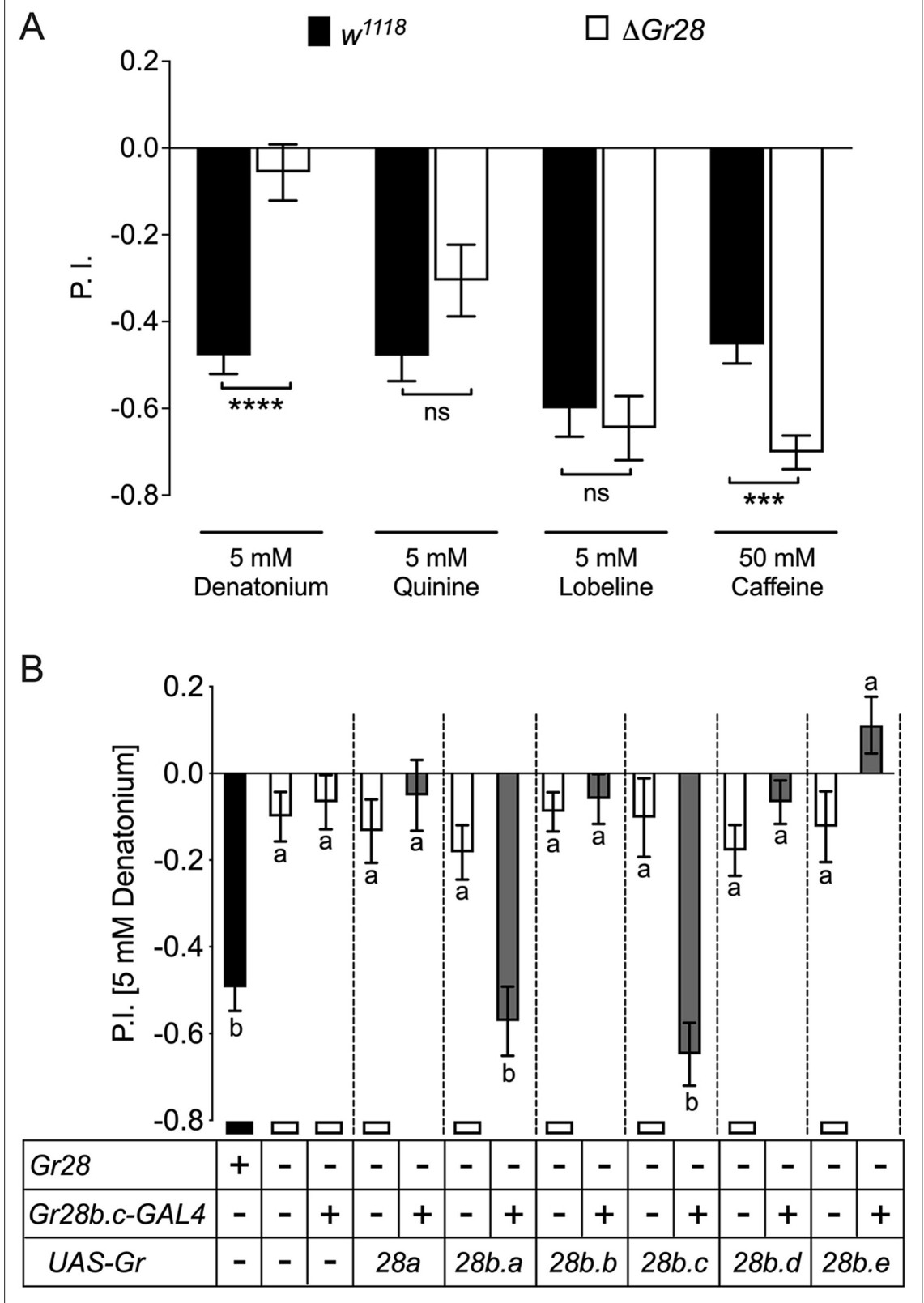

**Figure 4.** Role of individual *Gr28* genes in bitter taste avoidance. (**A**) *Gr28* genes are required for sensing denatonium. Wild-type (*w^1118^*) larvae, but not *Gr28* mutant larvae (*w^1118^;ΔGr28/ΔGr28*), strongly avoid denatonium. In contrast, *w^1118^;ΔGr28/ΔGr28* larvae do not show significantly reduced avoidance to quinine, lobeline and caffeine. Each bar represents the mean ± SEM of preference index (P.I.) (n = 12–22 assays). Asterisks indicate a significant difference between *w^1118^; ΔGr28/ΔGr28* and *w^1118^* larvae (two-tailed, Mann–Whitney *U* test, ****p<0.0001, ***p<0.001, ns, not significant). (**B**) Single

*Figure 4 continued on next page*

*Figure 4 continued*

*Gr28b* genes can rescue avoidance response to denatonium when expressed in *Gr28b.c* neurons of *Gr28* mutant larvae. The behavior of $w^{1118}$;*ΔGr28/ΔGr28* larvae expressing *UAS-Gr28* transgenes under control of the *Gr28b.c-GAL4* driver was compared to *Gr28+* control ($w^{1118}$, black bar), and three *Gr28* mutant controls (*ΔGr28, ΔGr28* plus driver and *ΔGr28* plus respective *UAS-Gr28* transgene, white bar) using Kruskal–Wallis test by ranks with Dunn's multiple comparison tests ($p<0.05$). Each bar represents the mean ± SEM of P.I. (n = 11–22 assay). Bars with different letters are significantly different. Dashed lines delineate groups for ANOVA. Fly genotypes: wild-type: $w^{1118}$ (black), mutants: $w^{1118}$;*ΔGr28/ΔGr28*, $w^{1118}$; *ΔGr28/ΔGr28 Gr28b.c-GAL4*, $w^{1118}$;*ΔGr28/ΔGr28; UAS-Gr28* (indicated *Gr28* genes)/+ and $w^{1118}$; *ΔGr28/ΔGr28 UAS-GCaMP6m; UAS-Gr28* (for *Gr28b.b* or *Gr28b.c* genes)/+ (white), rescues: $w^{1118}$; *ΔGr28/ΔGr28 Gr28b.c-GAL4; UAS-Gr28* (indicated *Gr28* genes)/+ and $w^{1118}$; *ΔGr28 UAS-GCaMP6m/ΔGr28 Gr28b.c-GAL4; UAS-Gr28* (for *Gr28b.b* or *Gr28b.c* genes)/ + (gray).

The online version of this article includes the following source data and figure supplement(s) for figure 4:

**Source data 1.** Taste response to bitter compounds of *Gr28* mutant larvae.

**Figure supplement 1.** *Gr66a* is necessary for caffeine avoidance of larvae.

**Figure supplement 1—source data 1.** Taste response to bitter compounds of *Gr66a* mutant larvae.

ribonucleosides and RNA, which are appetitive nutrients essential for larvae and detected by *Gr28a* neurons (*Mishra et al., 2018*). Indeed, RNA has been found to be an appetitive taste ligand across many dipteran insects, including mosquitoes, and we showed that Gr28 homologs of both *A. aegypti* and *A. gambiae* can rescue the preference for RNA and ribose when expressed in *Gr28a* neurons of *ΔGr28* mutant larvae (*Fujii et al., 2023*).

Previous studies in adult *Drosophila* have shown that members of conserved Gr protein families such as the carbon dioxide receptors (Gr21a and Gr63a) (*Jones et al., 2007*; *Kwon et al., 2007*) or the receptors for sweet taste encoded by the eight sugar *Gr* genes (*Fujii et al., 2015*) are largely co-expressed in one type of neuron in the fly's taste organs. For example, with the exception of *Gr5a* (see below), sugar *Gr* genes are only expressed in a single GRN (the 'sweet' neuron) of each taste sensilla, and activation of these 'sweet' neurons by sugars requires the function of at least two of the eight sugar *Gr* genes (*Dahanukar et al., 2007*; *Yavuz et al., 2014*; Fujii et al., unpublished). Similarly, the approximately 33 putative bitter taste receptors, which comprise several small conserved subfamilies (*Robertson et al., 2003*), as well as individual *Gr* genes with little overall similarity to one another, are partially co-expressed in the bitter GRN of each taste sensilla (*Weiss et al., 2011*). Molecular genetic studies combined with electrophysiological recordings have shown that at least three different Gr subunits are required to constitute functional receptor complexes that can sense a bitter compound (*Shim et al., 2015*). We note that two rare exceptions to the heteromeric nature of taste receptor complexes exist, namely the RNA receptor Gr28a and the fructose receptor Gr43a, which have been proposed to function as homomultimeric complexes (*Mishra et al., 2018*; *Mishra et al., 2013*). Cryo-EM structural analysis of the conserved insect olfactory receptor co-receptor (ORCO) suggests that insect odorant receptors form tetramers (*Butterwick et al., 2018*), and biochemical characterization and comparative modeling of BmGr9, the *Bombyx mori* homolog of the *Drosophila* Gr43a fructose receptor, supports such structures for Gr proteins as well (*Morinaga et al., 2022*).

## Distinct functions are mediated by small set of GRNs expressing specific Gr28 subunits

Our expression analysis of the bitter taste receptor gene *Gr66a* and the *Gr28* genes in larvae is consistent with earlier studies, despite some small variation in neuron number (*Choi et al., 2016*; *Kwon et al., 2011*), which is likely due to the use of different *GAL4* driver lines and/or variability in expression levels. Importantly, all *Gr28b* genes are co-expressed with the bitter taste receptor gene *Gr66a* (*Figure 1*) and probably several other putative bitter *Gr* genes (*Kwon et al., 2011*; *Rist and Thum, 2017*), while *Gr28a* is found in a largely, but not entirely, distinct set of GRNs. Whether and what kind of *Gr* genes might be co-expressed with *Gr28a* in *Gr28a^only^* GRNs will require more in-depth expression studies and might shed light on other receptors involved in appetitive behaviors of larvae.

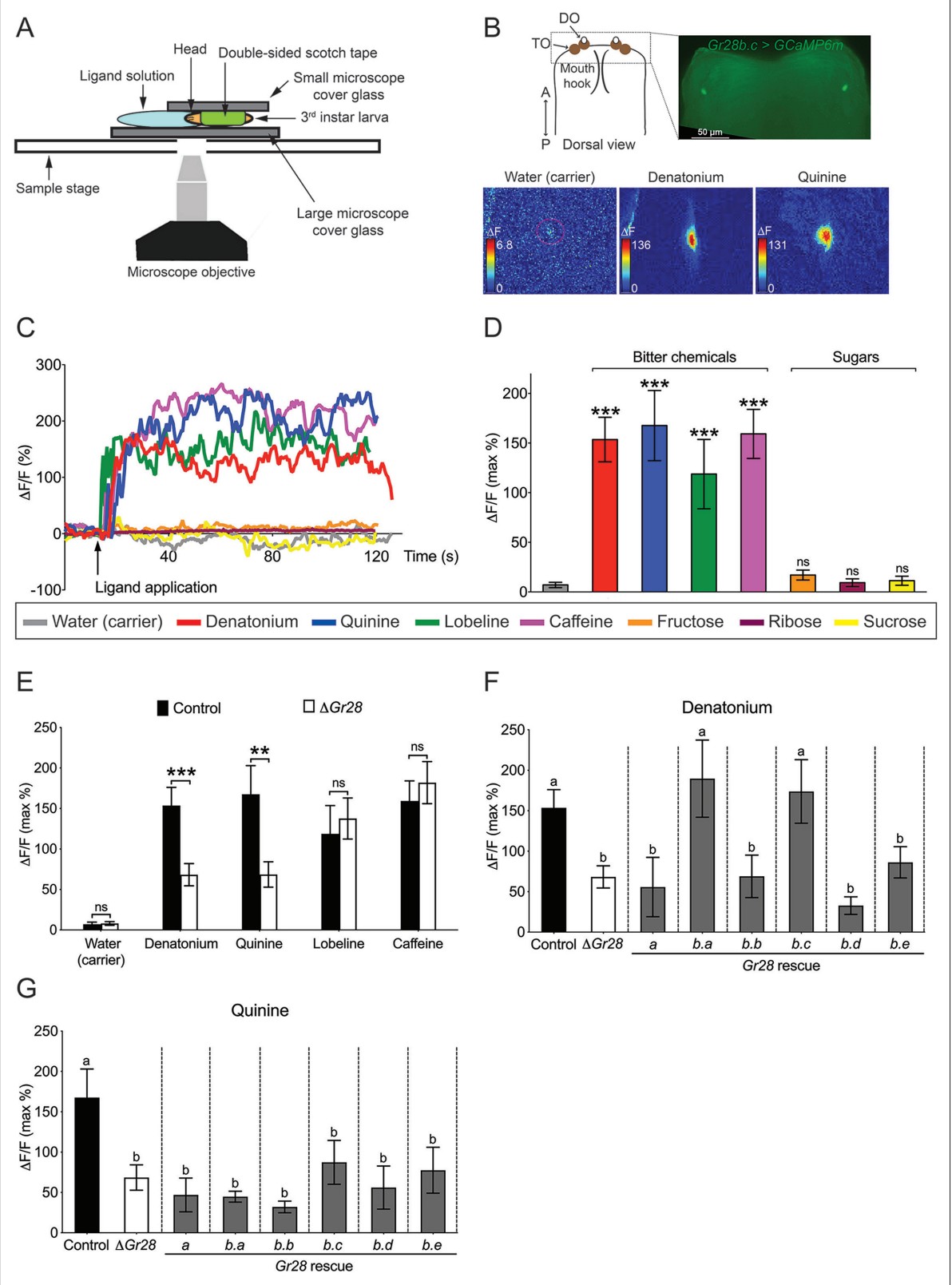

**Figure 5.** Cellular Ca²⁺ responses of *Gr28b.c* gustatory receptor neurons (GRNs) to select bitter compounds requires Gr28b.a or Gr28b.c. (**A**) Diagram of Ca²⁺ imaging experimental set up. (**B**) Representative still images of Ca²⁺ response in the *Gr28b.c* expressing GRN of the TOG. Ca²⁺ responses of the *Gr28b.c* GRNs upon stimulation with indicated ligands. ΔF indicates the changes in fluorescence light intensity of the cell body after ligand application. (**C, D**) Representative traces (**C**) and quantified Ca²⁺ responses (**D**) of the *Gr28b.c* GRNs after stimulation with indicated ligands. Fly genotype: *w^1118^;*

*Figure 5 continued on next page*

*Figure 5 continued*

*Gr28b.c-GAL4/UAS-GCaMP6m*. Each bar represents the mean ± SEM of Ca²⁺ imaging with 12–16 larvae. Asterisks indicate a significant difference between carrier (water) and indicated ligands (two-tailed, Mann–Whitney *U* test, ***p<0.001, ns, not significant). (**E**) Neurons of larvae lacking the *Gr28* genes exhibit significantly reduced responses to denatonium and quinine. *Gr28b.c*-expressing GRNs in the TOG of *Gr28* mutant larvae (Δ*Gr28*) have significantly reduced Ca²⁺ responses to denatonium and quinine but not to lobeline or caffeine when compared to *Gr28b.c*-expressing GRNs of wild-type controls. Larvae genotypes: *Gr28⁺* control (black bar): *w^1118^*; *Gr28b.c-GAL4/UAS-GCaMP6m*. Δ*Gr28* control (white bar): *w^1118^*; Δ*Gr28 Gr28b.c-GAL4/ ΔGr28 UAS-GCaMP6m*. Each bar represents the mean ± SEM with 13–16 larvae. Asterisks indicate a significant difference between *Gr28⁺* and Δ*Gr28* larvae (two-tailed, Mann–Whitney *U* test, ***p<0.001, **p<0.01; ns, not significant). (**F, G**) *Gr28b.c* or *Gr28b.a* transgenes rescue denatonium responses in *Gr28b.c-GAL4* neurons of Δ*Gr28* larvae. Expression of *Gr28b.c* or *Gr28b.a* is under control of *Gr28b.c-GAL4* restores responses to denatonium, but not quinine in TOG GRNs of Δ*Gr28* larvae. Each bar represents the mean ± SEM of Ca²⁺ imaging with 12–17 larvae. The Ca²⁺ responses of *Gr28* mutant larvae expressing *UAS-Gr28* transgenes under *Gr28b.c-GAL4* driver is compared to *Gr28⁺* (black) and Δ*Gr28* (white) controls using Kruskal–Wallis test by ranks with Dunn's multiple comparison tests (p<0.05). Bars with different letters are significantly different. Dashed lines delineate groups for ANOVA. Fly genotypes: *Gr28⁺* control (black bar): *w^1118^*; *Gr28b.c-GAL4/UAS-GCaMP6m*. Δ*Gr28* control (white bar): *w^1118^*; Δ*Gr28 Gr28b.c-GAL4/ΔGr28 UAS-GCaMP6m*. *Gr28* rescues (gray bar): *w^1118^*; Δ*Gr28 Gr28b.c-GAL4/ΔGr28 UAS-GCaMP6m; UAS-Gr28* (indicated Gr28 genes)/+. Concentration of ligands was 100 mM for sugars, 50 mM for caffeine, and 5 mM for denatonium, quinine, and lobeline.

The online version of this article includes the following source data for figure 5:

**Source data 1.** Ca²⁺ imaging experiments with *Gr28b.c* GRNs in the TOG.

A key finding of the work presented here is the observation that *Gr28a^only^* and *Gr28b.c* neurons dictate distinct behavioral programs, the former representing an ensemble of neurons that instruct larvae to 'go toward' a chemical source and consume it, while the latter do the opposite (*Figure 2*). This observation is reminiscent of a seminal study by Troemel and colleagues in the *Caenorhabditis elegans* chemosensory system, who reported that the valence of a chemical compound is dependent on the identity of a neuron, and not the identity of the molecular receptor the neuron expresses (*Troemel et al., 1997*). The number of 'go-away' GRNs in *Drosophila* larvae co-expressing *Gr28b.c* and *Gr66a* is remarkably small, consisting of only two pairs, one in the TO and the other in the DPS/ VPS. It seems likely that this is the smallest, minimal subset of neurons sufficient to induce avoidance behavior as expression of VR1 in only the TO pair (under the control of either *Gr28b.a-GAL4* or *Gr28b.e-GAL4*) has no behavioral effect when challenged with capsaicin. The 'go-to' neurons are characterized by expression of *Gr28a* and represent a slightly larger set of four GRN pairs (*Gr28a^only^* GRNs) (*Figure 7*) . Thus, the minimal requirement to induce 'go-to' and 'go-away' behavior is defined by distinct sets of GRNs, and each appears to be composed of neurons located in both external and the internal taste organs. Co-expression of the RNA taste receptor Gr28a in the DPS/VPS GRN essential for bitter taste (*Figures 2B and 3*) raises interesting questions about additional functions for Gr28a in bitter taste. We note that the sweet taste receptor Gr5a, a subunit of a multimeric trehalose receptor, is also expressed in non-sweet neurons of unknown function (*Fujii et al., 2015*).

## Functional redundancy of taste receptors

Both behavioral analyses and Ca²⁺ imaging experiments implicate at least one Gr28b protein as an essential component of a denatonium receptor complex as Δ*Gr28* larvae exhibit total loss of avoidance (*Figure 4*) and respective GRNs fail to elicit a response upon exposure to this chemical (*Figure 3*). What the precise composition of that complex is remains to be determined, but recovery of denatonium responses by expressing either Gr28b.c and/or Gr28b.a indicates that either one of these (or possibly both) is an essential subunit, in addition to other Grs expressed in this GRNs, such as Gr22a and Gr59c (*Choi et al., 2020*; *Rist and Thum, 2017*), while Gr66a is unlikely to be part of such a complex based on our behavioral analysis (*Figure 4—figure supplement 1*).

Since only Gr28b.a and Gr28b.c can rescue denatonium responses in *Gr28b.c* GRNs of Δ*Gr28* mutant larvae, sequence comparison between the unique N-terminal halves of the Gr28 proteins comprising the first four transmembrane domains and the extracellular loops 1 and 2 might provide insights as to possible residues important for ligand recognition. When interrogating these regions,

```
                                                  IL1
Gr28b.a  ----------------MIRCGLDIFRGCRGRFRYWLSARDCYDSISLMVAIAFALGITPFLVR   47
Gr28b.c  MDIEMAKEPVNPTDTPDIEVTPGLCQPLRRRFRRFVTAKQLYECLRPVFHVTYIHGLTSFYIS   63
Gr28b.b  -----------------------MSALRRVRKYFISSQVYEALRPLFFLTFLYGLTPFHVV   38
Gr28b.d  -------------------------MSFYFCEIFKPRDAFGAEQTLLLYTYLLGLTPFRL-   35
Gr28b.e  -------------------------MWLLRRSVGKSGNRPHDVYTCYRLTIFMALCLGIVPYYVS   40
Gr28a    -------------------------MAFKLWERFSQADNVFQALRPLTFIS-LLGLAPFRLN   36
                                       :  :  .       :     *:. : :

                             TM1                    EL1           TM2
Gr28b.a  -RNALGENSLEQSWYGFLNAIFRWLLLAYCYSYI-NLRNESLIGYFMRNHVSQISTRVHDVGG   108
Gr28b.c  CDTKTGKKAIKKTIFGYINGIMHIAMFVFAYSLTIYNNCESVASYFFRSRITYFGDLMQIVSG   126
Gr28b.b  -RRKMGESYLKMSCFGVFNIFIYICLCGFCYIS-SLRQGESIVGYFFRTEISTIGDRLQIFNG   99
Gr28b.d  -RGQAGERQFHLSKIGYLNAFLQLSFFSYCFLAA-LIEQQSIVGYFFKSEISQMGDSLQKFIG   96
Gr28b.e  -ISSEGRGKLTSSYIGYINIIIRMAIYMVNSFYG-AVNRDTLMSNFFLTDISNVIDALQKING   101
Gr28a    L---NPRKEVQTSKFSFFAGIVHFLFFVLCFGIS-VKEGDSIIGYFFQTNITRFSDGTLRLTG   95
             .   . :  . :  :. :          .  :::  . *:  . :: .      . *

                                  IL2            TM3
Gr28b.a  IIAAVFTFILPLLLRKYFLKSVKNMVQVDTQ-LERLRSPVNFNTVVGQVVLVILAVVLLDTVL   170
Gr28b.c  FIGVTVIYLTAFVPNHRLERCLQKFHTMDVQ-LQTVGVKIMYSKVLRFSYMVLISMFLVNVLF   188
Gr28b.b  LIAGAVIYTSAILKRCKLLGTLTILHSLDTN-FSNIGVRVKYSRIFRYSLLVLIFKLLILGVY   161
Gr28b.d  MTGMSILFLCSSIRVRLLIHIWDRISYIDDR-FLNLGVCFNYPAIMRLRLLQIFLINGVQLGY   158
Gr28b.e  MLGIFAILLISLLNRKELLKLLATFDRLETEAFPRVGVAMHQVAANKKMNRLVIILVGSMVAY   164
Gr28a    ILAMSTIFGFAMFKRQRLVSIIQNNIVVDEI-FVRLGMKLDYRRILLSSFLISLGMLLFNVIY   157
             :  .         :        ::   :   :     .                 :

                          EL2                TM4                  IL3
Gr28b.a  LTTGLVCLAKME-VYASWQLTFIFVYELLAISITICMFCLMTRTVQRRITCLHKVLKNLAHQW   232
Gr28b.c  TGGTFSVLYSSE-VAPTMALHFTFLIQHTVIAIAIALFSCFTYLVEMRLVMVNK   241
Gr28b.b  FVGVFRLLVSLD-VTPSFCVCMTFFLQHSVVSIAICLFCVIAFSFERRLSIINQ   214
Gr28b.d  LISSNWMLLGND-VRPIYTAIVAFYVPQIFLLSIVMLFNATLHRLWQHFTVLNQ   211
Gr28b.e  ITCSFLMISLRDTTTFSISAVISFFSPHFIVCAVSFLAGNVMIKLRIYLSALNE   218
Gr28a    LCVSYSLLVSAT-ISPSFVTFTTFALPHINTSLMVFKFLCTTDLARSRFSMLNEILQDILDAH   219
             :            *           :                 :      :*:::  .

Gr28b.a  DT----R---SLKA-VNQKQRS-------LQCLDSFSMYTIVTKDPAEIIQESMEIHHLICEA   280
Gr28a    IEQLSALELSPMHSVVNHRRYSHRLRNLISTPMKRYSVTSVIRLNPEYAIKQVSNIHNLLCDI   282
             :::  **::: *          :. :*: ::: :*    *::  :**:*:*:

                         TM5              EL3              TM6
Gr28b.a  AATANKYFTYQLLTIISIAFLIIVFDAYYVLETLLGKSKRESKFKTVEFVTFFSCQMILYLIA   343
Gr28a    CQTIEEYFTYPLLGIIAISFLFILFDDFYILEAILN-PKRLDVFEADEFFAFFLMQLIWYIVI   344
           . *  ::.:**** ** **:*:**:*:** :*:**::*.     **  . *:: **.:**   *:* *::

                                        IL4
Gr28b.a  IISIVEGSNRAIKKSEKTGGIVHSLLNKTKSAEVKEKLQQFSMQLMHLKINFTAAGLFNIDRT   406
Gr28a    IVLIVEGSSRTILHSSYTAAIVHKILNITDDPELRDRLFRLSLQLSHRKVLFTAAGLFRLDRT   407
           *: *****.*:* :*. *..***.:** *..  *::::* ::*:** * *: ******.:***

                  TM7              EL4
Gr28b.a  LYFTISGALTTYLIILLQFTSNSPNNGYGNGSSCCETFNNMTNHTL--   452
Gr28a    LIFTITGAATCYLIILIQFRFTHHMDDTSSNS-----TNNLHSIHLGD   450
           * ***:** * *****:**  .   :. ...*    **: .  *
```

**Figure 6.** Amino acid alignment of the six Gr28 proteins. The sequence alignment was generated using Clustal Omega tool from ClustalW2 (https://www.ebi.ac.uk/Tools/msa/clustalo/). IL and EL indicate intracellular loop and extracellular loop, respectively. Note that the C terminal region starting at the IL3 is identical in the Gr28b proteins. Red highlighted letters indicate amino acids identical only in Gr28b.a and Gr28b.c. Green highlighted letters indicate amino acids conserved in Gr28b.a, Gr28b.c, and one other Gr28 protein. Asterisks below the sequences indicate residues identical in all Gr28

*Figure 6 continued on next page*

*Figure 6 continued*

proteins, colons indicate conserved residue (STA, NEQK, NHQK, NDEQ, QHRK, MILV, MILF, HY, FYW), and periods indicate moderately conserved residue (CSA, ATV, SAG, STNK, STPA, SGND, SNDEQK, NDEQHK, NEQHRK, FVLIM, HFY). TM1-7 indicate helical transmembrane segments predicted using HMMTOP 2.0 software.

only seven residues are identical between Gr28b.a and Gr28b.c (*Table 1* and *Figure 6*). Reducing the stringency requirement by allowing one of the remaining receptors to share the same residue, nine additional sites are identified. One experimental avenue to validate these residues as important sites contributing to denatonium binding might involve introduction of point mutations that converts respective amino acids of other Gr28b proteins into those found in Gr28b.a/Gr28b.c.

The role of Gr28b proteins in quinine detection is less clear, and the different phenotypes observed in behavioral experiments and $Ca^{2+}$ imaging suggest that at least two molecular types of quinine receptors exist in larvae. $Ca^{2+}$ imaging experiments implicate a role for multiple Gr28b subunits in a quinine receptor complex in the TOG neuron since single *Gr28b* genes cannot restore the loss of quinine response in *ΔGr28* larvae (*Figure 5E and G*). However, because *ΔGr28* mutant larvae still avoid quinine (*Figure 4*), at least one Gr28b-independent receptor must exist in one or several other GRNs, one of which is likely the *28b.c* GRN in the DPS/VPS, since expression of Kir2.1 in that neuron,

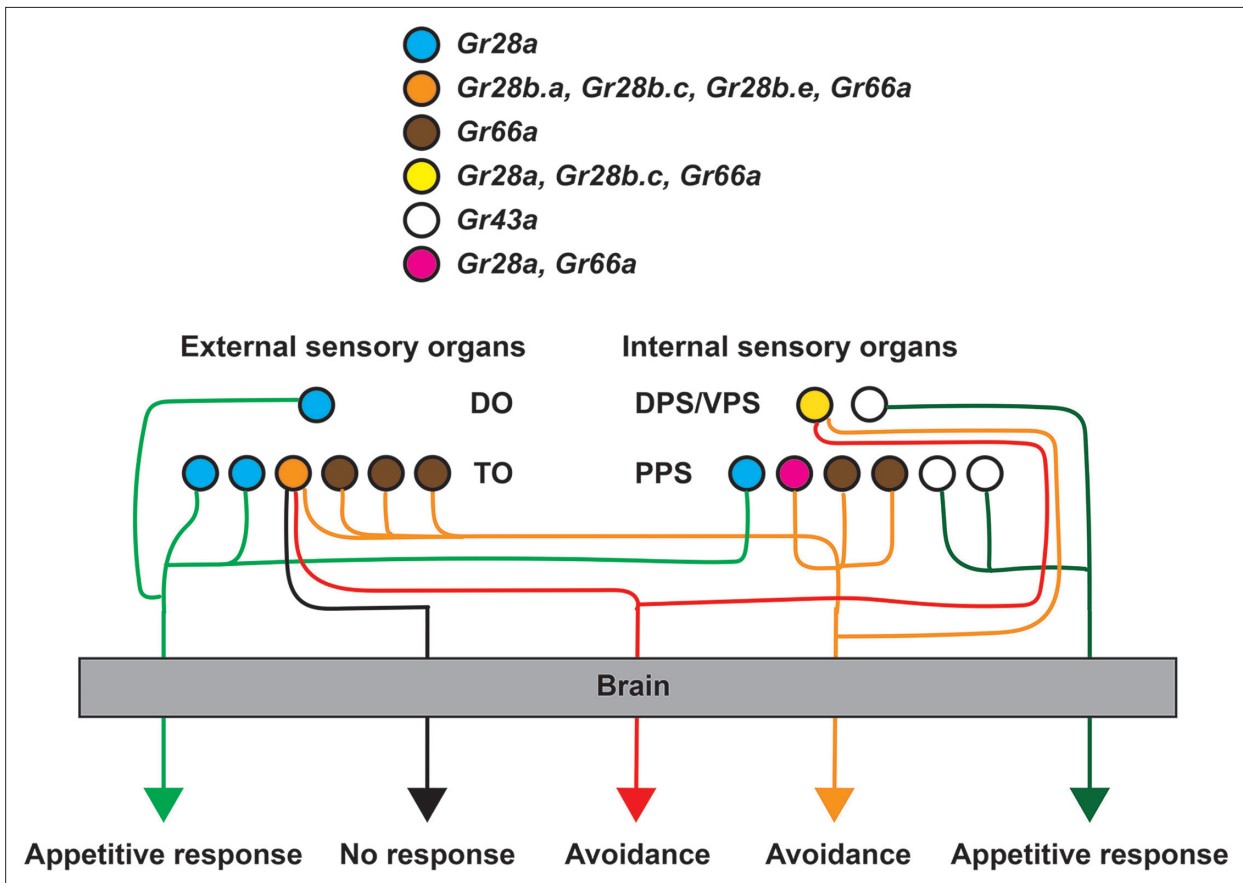

**Figure 7.** Role of different gustatory receptor neuron (GRN) subsets in taste behavior of larvae. GRNs sufficient for mediating avoidance behavior can be defined by *Gr28b.c-GAL4*, while GRNs sufficient for mediating appetitive behavior are defined by a subset of *Gr28a-GAL4* GRNs (*Gr28a$^{only}$* GRNs; see also *Figure 2B* and *Figure 2—figure supplement 1*). Note that each ensemble is composed of at least a pair of neurons located in the external taste organs and a pair of neurons in the internal taste organs. Also, a larger set of avoidance neurons (*Gr66a-GAL4*) might function independently of any *Gr28b.c* neurons, and one set of fructose sensing neurons (*Gr43a$^{GAL4}$*) distinct from *Gr28a-GAL4* GRNs mediates appetitive behavior.

**Table 1.** Conserved amino acids in the amino termini of Gr28b.a and Gr28b.c.

The seven amino acid residues identical in the amino-terminal region of Gr28b.c and Gr28b.a are shown in bold (residue number is taken from Gr28b.c). The nine additional amino acids also identical in one additional Gr28 proteins are also listed. These residues are considered potentially critical for recognition of denatonium since only Gr28b.a and Gr28b.c can rescue response to denatonium when expressed in *Gr28b.c* neurons of *ΔGr28/ΔGr28* mutant larvae.

| Location | Conserved in Gr28b.c/Gr28b.a | Other (if applicable) |
|---|---|---|
|  | E103 | Gr28b.b |
| EL1 | R111 | Gr28b.b |
| EL2 | **E200** | **None** |
|  | Y94 | Gr28b.b |
| TM1 | **S95** | **None** |
|  | **V124** | **None** |
| TM2 | I128 | Gr28b.b |
|  | **V170** | **None** |
| TM3 | V177 | Gr28b.b |
|  | **L207** | **None** |
|  | **F209** | **None** |
|  | I218 | Gr28a |
|  | I220 | Gr28b.b |
|  | I222 | Gr28b.b |
|  | T229 | Gr28a |
| TM4 | **V232** | none |

along with the one in the TOG, totally abolishes quinine avoidance (*Figure 3*). We note that functional redundancy is also observed in sweet taste receptors, where we found that different combinations of sugar *Gr* genes can restore responses to the same sugar when expressed in tarsal sweet GRNs of sugar blind flies (Fujii et al. unpublished).

# Materials and methods

**Key resources table**

| Reagent type (species) or resource | Designation | Source or reference | Identifiers | Additional information |
|---|---|---|---|---|
| Antibody | Anti-GFP (rabbit polyclonal) | Thermo Fisher Scientific | Cat# A6455, RRID:AB_221570 | IF (1:1000) |
| Antibody | Anti-mCD8 (rat monoclonal) | Thermo Fisher Scientific | Cat# MCD0800, RRID:AB_10392843 | IF (1:200) |
| Antibody | Anti-rabbit Alexa 488 (goat polyclonal) | Thermo Fisher Scientific | Cat# A11070, RRID:AB_2534114 | IF (1:500) |
| Antibody | Anti-rat Cy3 (goat polyclonal) | Jackson ImmunoResearch Laboratories Inc | Cat# 112-165-072, RRID:AB_2338248 | IF (1:300) |
| Chemical compound, drug | Caffeine | MilliporeSigma | C0750 |  |
| Chemical compound, drug | Capsaicin | MilliporeSigma | M2028 |  |
| Chemical compound, drug | Denatonium benzoate | MilliporeSigma | D5765 |  |
| Chemical compound, drug | Lobeline hydrochloride | MilliporeSigma | 141879 |  |

*Continued on next page*

*Continued*

| Reagent type (species) or resource | Designation | Source or reference | Identifiers | Additional information |
|---|---|---|---|---|
| Chemical compound, drug | D-(-)-ribose | MilliporeSigma | R7500 | |
| Chemical compound, drug | Quinine hydrochloride dihydrate | MilliporeSigma | Q1125 | |
| Chemical compound, drug | Fructose | Spectrum Chemical | F1092 | |
| Chemical compound, drug | Agarose | Apexbio | 20-102 | |
| Chemical compound, drug | Sucrose | Macron Fine Chemicals | 8360-06 | |
| Chemical compound, drug | Charcoal | J.T. Baker | 1560-01 | |
| Genetic reagent (*Drosophila melanogaster*) | $w^{1118}$ | Bloomington Drosophila Stock Center | BDSC: 3605; FLYB: FBst0003605 | |
| Genetic reagent (*D. melanogaster*) | Gr28a-GAL4 | **Thorne and Amrein, 2008** | FLYB: FBtp0056017 | FlyBase symbol: $w*$; $P\{Gr28a\text{-}GAL4.T\}SF36S$ |
| Genetic reagent (*D. melanogaster*) | Gr28a-GAL4 | **Thorne and Amrein, 2008** | FLYB: FBtp0056017 | FlyBase symbol: $w*$; $P\{Gr28a\text{-}GAL4.T\}SF36B1$ |
| Genetic reagent (*D. melanogaster*) | Gr28b.a-GAL4 | **Thorne and Amrein, 2008** | FLYB: FBtp0054526 | FlyBase symbol: $w*$; $P\{Gr28b.a\text{-}GAL4\}NT42aC51a$ |
| Genetic reagent (*D. melanogaster*) | Gr28b.c-GAL4 | **Thorne and Amrein, 2008** | FLYB: FBtp0054528 | FlyBase symbol: $w*$; $P\{Gr28b.c\text{-}GAL4\}NT21B1$ |
| Genetic reagent (*D. melanogaster*) | Gr28b.e-GAL4 | **Scott et al., 2001** | FLYB: FBtp0014672 | FlyBase symbol: $w*$; $P\{Gr28b.e\text{-}GAL4.4.245\}Gr28a3AII$ |
| Genetic reagent (*D. melanogaster*) | ΔGr28/ΔGr28 | **Mishra et al., 2018** | FLYB: FBab0049019 | FlyBase symbol: $w*$; $Df(2L)\Delta Gr28$ |
| Genetic reagent (*D. melanogaster*) | Gr66a-GAL4 | **Scott et al., 2001** | FLYB: FBtp0014661 | FlyBase symbol: $w*$; $P\{Gr66C1\text{-}GAL4.3.153\}$ |
| Genetic reagent (*D. melanogaster*) | UAS-Gr28a | **Ni et al., 2013** | FLYB: FBal0344045 | FlyBase symbol: $w*$; $P\{UAS\text{-}Gr28a.G\}attP2$ |
| Genetic reagent (*D. melanogaster*) | UAS-Gr28b.a | **Ni et al., 2013** | FLYB: FBal0291410 | FlyBase symbol: $w*$; $P\{UAS\text{-}Gr28b.A\}attP2$ |
| Genetic reagent (*D. melanogaster*) | UAS-Gr28b.b | **Ni et al., 2013** | FLYB: FBal0291412 | FlyBase symbol: $w*$; $P\{UAS\text{-}Gr28b.B\}attP2$ |
| Genetic reagent (*D. melanogaster*) | UAS-Gr28b.c | **Ni et al., 2013** | FLYB: FBal0291411 | FlyBase symbol: $w*$; $P\{UAS\text{-}Gr28b.C\}attP2$ |
| Genetic reagent (*D. melanogaster*) | UAS-Gr28b.d | **Ni et al., 2013** | FLYB: FBal0291409 | FlyBase symbol: $w*$; $P\{UAS\text{-}Gr28b.D\}attP2$ |
| Genetic reagent (*D. melanogaster*) | UAS-Gr28b.e | **Ni et al., 2013** | FLYB: FBal0291408 | FlyBase symbol: $w*$; $P\{UAS\text{-}Gr28b.E\}attP2$ |
| Genetic reagent (*D. melanogaster*) | $Gr43a^{GAL4}$ | **Miyamoto et al., 2012** | BDSC:93447; FLYB:FBst0093447 | FlyBase symbol: $w^{1118}$; $Ti\{GAL4\}Gr43a^{GAL4}$ |
| Genetic reagent (*D. melanogaster*) | UAS-VR1E600K | **Marella et al., 2006** | FLYB: FBal0215202 | FlyBase symbol: $w^{1118}$; $P\{UAS\text{-}VR1E600K\}$ |
| Genetic reagent (*D. melanogaster*) | lexAop-rCD2:GFP | **Lai and Lee, 2006** | FLYB: FBst0066687 | FlyBase symbol: $w*$; $P\{lexAop\text{-}rCD2\text{-}GFP\}$ |
| Genetic reagent (*D. melanogaster*) | UAS-mCD8:RFP | Bloomington Drosophila Stock Center | BDSC: 32220; FLYB: FBti0131987 | FlyBase symbol: $y^1w*$;$P\{10XUAS\text{-}IVS\text{-}mCD8::RFP\}su(Hw)attP8$ |
| Genetic reagent (*D. melanogaster*) | UAS-GCaMP6m | Bloomington Drosophila Stock Center | BDSC: 42748; FLYB: FBti0151346 | FlyBase symbol: $w^{1118}$; $P\{20XUAS\text{-}IVS\text{-}GCaMP6m\}attP40$ |
| Genetic reagent (*D. melanogaster*) | UAS-Kir2.1-GFP | **Baines et al., 2001**; **Paradis et al., 2001** | FLYB: FBst0006596 | FlyBase symbol: $w*$; $P\{UAS\text{-}Hsap\backslash KCNJ2.EGFP\}1$ |

*Continued on next page*

*Continued*

| Reagent type (species) or resource | Designation | Source or reference | Identifiers | Additional information |
|---|---|---|---|---|
| Genetic reagent (*D. melanogaster*) | *Gr66a-LexA* | **Thistle et al., 2012** | BDSC: 93024; FLYB: FBst0093024 | FlyBase symbol: *w[1118]; P{Gr66a-lexA.S}2;TM2/TM6B* |
| Genetic reagent (*D. melanogaster*) | *lexAop-GAL80* | **Thistle et al., 2012** | FLYB: FBtp0079728 | FlyBase symbol: *w[1118]; P{lexAop-GAL80. T}* |
| Genetic reagent (*D. melanogaster*) | *Gr28b.c-LexA* | This paper | | FlyBase symbol: *w[1118];P{Gr28b.c-LexA}#8* |
| Sequence-based reagent | Gr28b.c_F | This paper | PCR primers | 5′-AATCTAGGTACCCCGGCTGCTCGTCTCCCTGGATGT-3′ |
| Sequence-based reagent | Gr28b.c_R | This paper | PCR primers | 5′-CGTCAAACTAGTGACCGCTTCGTTTGAGCTTCAACC-3′ |
| Recombinant DNA reagent | LexA vector CMC105 (plasmid) | This paper **Larsson et al., 2004** | | Insect expression vector |
| Software, algorithm | NIS-Elements | Nikon | N/A | |
| Software, algorithm | Prism software 10.1.0 (264) | GraphPad Software | N/A | |
| Software, algorithm | Adobe pPhotoshop 2022 | Adobe | N/A | |
| Other | Normal goat serum | SouthernBiotech | Cat# 0060-01 | IF (5%) 'Materials and methods' |
| Other | Nikon Eclipse Ti inverted microscope | Nikon | N/A | 'Materials and methods' |
| Other | Nikon A1R confocal microscope system | Nikon | N/A | 'Materials and methods' |
| Other | PertriPetri dish, 60 × 15 mm | Falcon | REF353004 | 'Materials and methods' |
| Other | Microscope cover glass, 24 × 50 mm | VWR | 16004-098 | 'Materials and methods' |
| Other | Microscope cover glass, 12CIR-1 | Thermo Fisher Scientific | 1254580 | 'Materials and methods' |

## *Drosophila* stocks

Flies were maintained on standard corn meal food in plastic vials under a 12 hr light/dark cycle at 25°C. The *w[1118]* strain (Bloomington Drosophila Stock Center, number 3605) was used as a wild-type control. Fly strains used: *Gr28a-GAL4(SF36S)* for **Figures 1 and 2B**, and *SF36E1* for **Figure 2B** and **Figure 2—figure supplement 1**, *Gr28b.a-GAL4(NT42aC51a)*, *Gr28b.c-GAL4(NT21B1)* (**Thorne and Amrein, 2008**); *Gr28b.e-GAL4(Gr28a3AII)* and *Gr66a-GAL4* (**Scott et al., 2001**); *ΔGr28(54B3)* (**Mishra et al., 2018**); *UAS-Gr28a*, *UAS-Gr28b.a*, *UAS-Gr28b.b*, *UAS-Gr28b.c*, *UAS-Gr28b.d*, and *UAS-Gr28b.e* (**Ni et al., 2013**); *Gr43a[GAL4]* (**Miyamoto et al., 2012**); *Gr66a-LexA* and *lexAop-GAL80* (**Thistle et al., 2012**); *UAS-VR1E600K* (**Marella et al., 2006**); *UAS-Kir2.1-GFP* (**Baines et al., 2001**; **Paradis et al., 2001**); *lexAop-rCD2:GFP* (**Lai and Lee, 2006**), *UAS-GCaMP6m*, *UAS-mCD8:RFP* and *Gr66a[ex83]* (Bloomington Drosophila Stock Center, numbers 42748, 32220, and 35528); *Gr28b.c-LexA*(#8).

## Chemicals

Caffeine (Cat# C0750), capsaicin (Cat# M2028), denatonium benzoate (Cat# D5765), lobeline hydrochloride (Cat# 141879), D-(-)-ribose (Cat# R7500), and quinine hydrochloride dihydrate (Cat# Q1125) were purchased from MilliporeSigma, with a purity of >95%. Fructose (Cat# F1092) and agarose (Cat# 20-102) were purchased from Spectrum chemical and Apexbio, respectively. Sucrose (mfr. no. 8360-06) and charcoal (Cat# 1560-01) were purchased from Macron Fine Chemicals and J.T. Baker, respectively. A stock solution for capsaicin (20 mM) was prepared in 70% ethanol and stored at 4°C protected from light for up to 1 y. Stock solutions for bitter chemicals were prepared in Millipore Q water and stored at –20°C. Stock solutions for sugars were prepared in Millipore Q water and stored at 4°C for up to 1 mo. A stock solution for ribose was treated with charcoal (10% of the weight of ribose used for stock

solution) overnight at 4°C and sterile-filtrated (0.45 μm) to remove unrelated odor. Stock solutions were diluted to the final concentration using Millipore Q water prior to each experiment.

## Immunofluorescence

Immunofluorescence of larval heads was performed based on the protocol described in Croset and colleagues (*Croset et al., 2016*) with minor modification. Heads of third-instar larvae were dissected using microscissors in phosphate-buffered saline (PBS) and immediately fixed in PBS with 4% paraformaldehyde for 1 hr at 4°C. They were washed six times in washing buffer (PBS with 0.1% Triton X-100) for 20 min and blocked for 1 hr in washing buffer containing 5% heat-inactivated goat serum (SouthernBiotech, Cat# 0060-01), followed by incubation with the primary antibodies (rabbit anti-GFP, 1:1000 dilution; rat anti-mCD8, 1:200 dilution, Thermo Fisher Scientific) at 4°C overnight. The next day, heads were washed six times for 20 min in washing buffer and blocked in washing buffer containing 5% heat-inactivated goat serum for 1 hr, followed by incubation with the secondary antibodies (goat anti-rabbit Alexa 488, 1:500 dilution, Thermo Fisher Scientific; goat anti-rat Cy3, 1:300 dilution, Jackson ImmunoResearch Laboratories Inc) at 4°C overnight. Finally, heads were washed six times in washing buffer for 20 min each at room temperature under gentle agitation. Heads were then mounted with VectaShield (Vector Lab, Cat# H-1200) on a microscope slide and images were obtained using a Nikon A1R confocal microscope system. Adobe Photoshop 2022 was used further to process images.

## Larval two-choice preference assay

Two-choice preference assay of larvae was conducted as described in *Mishra et al., 2013* with minor modifications. Flies were placed on standard corn meal food in plastic vials and allowed to lay eggs for 24 hr under a 12 hr light/dark cycle at 25°C. Flies were removed from food vials and feeding-stage third-instar larvae were collected. Agarose food dishes for two-choice preference assay were prepared just prior each experiment as follows: Petri dishes (60 × 15 mm, Falcon, Cat# REF353004) with two halves marked on the bottom were filled with melted plain 1% agarose or 1% agarose containing 1.75% ethanol (for capsaicin preference). After the agarose solidified, one half was removed and replaced with 1% agarose solution containing taste ligands (capsaicin or bitter compound). For each experiment, 15 larvae from food vials were briefly rinsed twice with Millipore Q water and placed along the middle separating pure and ligand containing agarose. After 16 min, images were taken for record keeping and used to calculate larval preference indices. Larvae that crawled onto the wall of a dish or dug in the agarose were excluded. The preference index (P.I.) was calculated as follow: PI = $(N_{tastant} − N_{plain})/N_{Total}$, whereby N is the number of larvae in the tastant sector, the plain agarose sector, and the total number, respectively. Positive values indicate a preference for capsaicin or bitter compound while negative values indicate repulsion (avoidance).

## Calcium imaging

Calcium imaging was performed in *Gr28b.c* GRNs expressed in the terminal organ of feeding-stage, third-instar larvae, reared as described for the larval two-choice preference assay. For each experiment, larvae from food vials were briefly rinsed twice with Millipore Q water and were mounted dorsally on a large microscope cover glass (24 × 50 mm, VWR, Cat# 16004-098) using double-sided scotch tape and covered with a small microscope cover glass (12CIR-1, Thermo Fisher Scientific, Cat# 1254580). Millipore Q water (40 μl) was applied to the tip of the larval head, and the preparation was placed on the stage of a Nikon eclipse Ti inverted microscope. Images were obtained every 500 ms, starting 15 s before application and ending 105 s after ligand application. Each recording was initiated by applying water (40 μl) to set a baseline. The first ligand solution (40 μl of bitter chemical or sugar) was applied thereafter, followed by five washes with carrier (100 μl of water). After a 3 min pause to allow the preparation to recalibrate, a second ligand solution (40 μl bitter chemical or sugar) was applied. To assure validity in experiments with *Gr28* mutants and rescues, each recording was concluded with application of caffeine, and recordings were included only if caffeine generated a positive response. Baseline fluorescence, which was determined from the average of five frame measurements from a region next to the cell immediately before ligand application, was subtracted from the actual measurements. ΔF/F (%) = (fluorescence light intensity of the cell body – baseline/baseline) × 100. ΔF/F (max %) is the maximum value within 40 s after ligand application.

## Generation of transgenic *Gr28b.c-LexA* flies

To generate the *Gr28b.c-LexA* driver, a 1.3 kb DNA fragment immediately upstream of the *Gr28b.c* start codon was amplified from *w^1118* flies using a forward (5'-AATCTA<u>GGTACC</u>CCGGCTGCTCGTCTCC CTGGATGT-3') and a reverse (5'- CGTCAA<u>ACTAGT</u>GACCGCTTCGTTTGAGCTTCAACC-3') primer. *Acc65I* and *SpeI* sites included in the primer sequence (underlined) were incorporated such that the amplified fragment was amenable to directional cloning into the LexA vector CMC105 (*Larsson et al., 2004*). The clone chosen was confirmed by DNA sequence analysis. Transgenic flies were generated by standard P-element transformation of *w^1118* embryos (Rainbow Transgenic Flies Inc, Camarillo, CA).

## Statistical analysis

Statistical analyses were conducted using Prism software 9.5.1 (GraphPad Software). Larval two-choice preference assay and $Ca^{2+}$ imaging data were analyzed for normal distribution using D'Agostino–Pearson omnibus and Shapiro–Wilk normality tests. When groups did not meet the assumption for normal distribution, nonparametric statistics was used. For comparison between multiple groups, one-way ANOVA or Kruskal–Wallis test by ranks (nonparametric one-way ANOVA) was performed to test for difference of mean or rank distribution. As a post hoc test, Bonferroni's or Dunn's (nonparametric) multiple comparison tests were employed to compare two specific groups. One-way ANOVA with Bonferroni's multiple comparison tests were used in *Figure 2B*. Kruskal–Wallis test by ranks with Dunn's multiple comparison tests were used in *Figures 3, 4B, and 5F and G*. For comparison between two groups, Mann–Whitney *U* test (nonparametric t -test, *Figures 4A, and 5D and E*, *Figure 2—figure supplement 1B*, and *Figure 4—figure supplement 1*) with two-tailed P- value were used. The sample size for larval two-choice preference assays and $Ca^{2+}$ imaging experiments werewas based on *Mishra et al., 2018*.

## Acknowledgements

We thank Tetsuya Miyamoto, Shinsuke Fujii, and Sheida Hedjazi for valuable suggestions throughout the duration of this project and Raquel Sitcheran for comments on the manuscript. We are grateful to Paul Garrity for the *UAS-Gr28* reporter strains and the Bloomington Stock Center for numerous *Drosophila* strains. This work was supported by NIH grants1 R01 DC018403, R21 DC015327, and R01GMDC05606 to H Amrein. Drs. Ahn and Amrein conceived the experiments, Dr. Ahn conducted all experiments, and Dr. Amrein wrote the paper.

## Additional information

### Funding

| Funder | Grant reference number | Author |
|---|---|---|
| National Institute on Deafness and Other Communication Disorders | R01 DC018403-01A1 | Hubert Amrein |
| National Institute on Deafness and Other Communication Disorders | 1R21 DC015327 | Hubert Amrein |
| National Institute on Deafness and Other Communication Disorders | 1RO1GMDC05606-01 | Hubert Amrein |

The funders had no role in study design, data collection and interpretation, or the decision to submit the work for publication.

### Author contributions

Ji-Eun Ahn, Conceptualization, Resources, Data curation, Formal analysis, Investigation; Hubert Amrein, Conceptualization, Funding acquisition, Writing - original draft, Project administration, Writing - review and editing

## Author ORCIDs

Ji-Eun Ahn  http://orcid.org/0009-0008-0947-4532
Hubert Amrein  http://orcid.org/0000-0001-8799-7250

Reviewer #1 (Public Review): https://doi.org/10.7554/eLife.89795.3.sa1
Reviewer #2 (Public Review): https://doi.org/10.7554/eLife.89795.3.sa2
Author Response https://doi.org/10.7554/eLife.89795.3.sa3

## Additional files

### Supplementary files

• MDAR checklist

### Data availability

All data generated or analysed during this study are included in the manuscript and supporting file. Source data files have been provided for Figures 2, 3, 4 and 5, and Figure 2—figure supplement 1 and Figure 4—figure supplement 1.

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
