## [Editor Report · eLife assessment]

This **valuable** study focuses on the role of the Gr28 family of insect chemoreceptors. Using the *Drosophila* larva, the authors show that taste neurons expressing different members of this family of bitter taste receptors trigger opposite behavior – attraction and repulsion. They establish the minimal bitter taste receptor subunit composition needed in these neurons to mediate the repulsion of bitter tastants. The evidence presented is **convincing**, using well-validated and controlled tools and experiments.

---

## [Referee Report · Reviewer #1 (Public Review)]

Ahn and Amrein characterize the expression of members of the Gr28 family of gustatory receptors in taste neurons in the *Drosophila melanogaster* larva, define the behaviorally-relevant ligands for these receptors, and use chemogenetic experiments to show, strikingly, that different neurons have opposite behavioral responses to the chemogenetic ligand. They go on to show what neurons need to be silenced to lose responses to bitters, and very nicely show what subunits of the Gr28 bitter receptors are necessary and sufficient for responses to bitters. This is a nice piece of work, rigorously carried out, that tackles the neurons and receptors that drive innate responses to tastants in Drosophila larvae.

The authors have revised the paper to address all of my recommendations. The new cartoons are extremely clear and I appreciate the more measured language when discussing the hypothetical structure and stoichiometry of the functional GR complex.

---

## [Referee Report · Reviewer #2 (Public Review)]

This study investigates how genes in the Gr28 family of gustatory receptors function in the taste system of *Drosophila* larvae. Gr28 genes are intriguing because they have been implicated in taste as well as other functions, such as sensing temperature and ultraviolet light. This study makes several new findings. First, the authors show that four Gr28 genes are expressed in putative taste neurons, and these neurons can be largely divided into subsets that express Gr28a versus Gr28bc. The authors then demonstrate that these two neuronal subsets drive opposing behaviors (attraction versus avoidance) when activated. The avoidance-promoting neurons respond to bitter compounds and are required for bitter avoidance, and Gr28bc and Gr28ba were specifically implicated in bitter detection in these cells. Together, these findings provide insight into the complexity of taste receptor expression and function in Drosophila, even within a single receptor subfamily.

The conclusions are well-supported by the experimental data. Strengths of the paper include the use of precise genetic tools, thorough analyses of expression patterns, carefully validated behavioral assays, and well-controlled functional imaging experiments. The role of Gr28bc neurons is more thoroughly explored than that of Gr28a neurons. However, a previous study from the same lab (Mishra et al., 2018) showed that Gr28a neurons detect RNA and ribose, which are attractive to larvae. Presumably this is the attractive response that is being recapitulated upon artificial activation of Gr28a neurons.

---

## [Author Response]

The following is the authors’ response to the original reviews.

General comments:

To reviewer 1 and 3: The following sentences below were added at the beginning of the result section to clarify that the Gr gene expression analysis was performed using bimodal expression systems and to provide a reference that these expression profiles can generally be expected to represent endogenous Gr expression.

"Note that this and all previous Gr expression studies were performed using bimodal expression systems, mostly GAL4/UAS, whereby Gr promotors driving GAL4 are assumed to faithfully reproduce expression of the respective Gr genes. Importantly, we analyzed two or more Gr28-GAL4 insertion lines for each transgene, and at least two generated the same expression profiles (Mishra et al., 2018; Thorne and Amrein, 2008) providing evidence that the drivers reflect a fairly accurate expression profile of respective endogenous genes."

Specific comments:

**Reviewer #1 (Recommendations For The Authors):**
The important chemogenetic behavioral data would benefit from a clearer presentation including a cartoon to explain what the behavior is and how it is scored. Figure 2 is the key figure in this paper and it would be helpful if the figure were reorganized to guide the non-expert reader to the key result. I recommend labeling the positive controls Gr43a as "sweet" and Gr66a as "bitter" and perhaps organize the presentation to have the negative control at the left, then Gr28ba that had no effect, then group Gr28a with Gr43a for positive valence and Gr28bc with Gr66a for negative valence. I'm not sure what the value is of showing both 0.1 mM and 0.5 mM capsaicin, the text does not explain. The experiment in Figure 2B is important but non-experts will not understand what is being done here - can the authors please provide a cartoon like those in Figure 1 showing what cells are being subjected to chemogenetics and how this differs from Figure 2A?

The reviewer is correct that much can be improved, which we hope to have accomplished with the modifications in Figure 2. We re-organized it to deliver the key result to non-expert readers in an easy way. We added cartoons both explaining how the two-choice preference assays were conducted and indicating which cells express UAS-VR1. The cartoon in Figure 1E and Figure 2A are now directly relatable and should clarify what cells express VR1 (in Figure 2). Positive and negative control experiments using Gr43aGAL4 (a GAL4 knock-in; Miyamoto et al., 2013) and Gr66a-GAL4 are highlighted in the Figure and mentioned upfront in the text to make clear to what the experimental larvae can be compared. We also excluded larvae responses to 0.5 mM capsaicin.

1. The AlphaFold ligand docking in Figure 8 is conducted with Gr28bc monomers, which are unlikely to be the in vivo relevant structure, given that the related OR/ORCO ancestor structures are tetramers. I recommend that this component of the paper either be removed entirely or that the authors redo the in silico work using the AlphaFold-Multimer package reported by Hassabis and Jumper in 2022 https://www.biorxiv.org/content/10.1101/2021.10.04.463034v2. It will be interesting to see what a tetramer structure looks like with the ligand.

We tried but were able to use the recommended package. Even if it were, the problem is that we do not know the partner of Gr28b.c. And while it is not clear whether and how extensive changes in the ligand binding pockets occur when using the monomer prediciton vs a multimer package, we followed the reviewer’s suggestion and removed the modeling from the manuscript.

Minor points:1. Line 80: I do not think it is biophysically or biochemically plausible that GRs and IRs would assemble into functional heteromeric channels and suggest that the authors either explain how that would work or remove this speculative comment.

We have removed this sentence.

1. Line 246-248: I would tone down the speculation about GR subunit composition - it's still too early days to understand the stoichiometry or the extent that any of the broadly expressed GRs is a co-receptor.

We did not indulge in the possible stoichiometry of Gr complexes, but merely mention that they are composed in general of two or more Gr subunits, for which clear genetic evidence exists: Up to three different putative bitter Gr genes are necessary to elicit responses to bitter compounds, and at least two putative sugar Gr genes are necessary to restore behavioral responses to any sweet tasting chemicals (sugars). Regardless, we have toned down the language, stating now:

“Given the multimeric nature of bitter taste receptors (Sung et al., 2017), one possibility is that the absence of a Gr subunit not required for the detection of denatonium (Gr66a) could favor formation of multimeric complexes containing Gr subunits that recognize this compound (Gr28b.a and/or Gr28b.c).”

1. Line 284: I don't think that co-expression necessarily means that GRs form heteromultimeric channels. It's equally possible that the cell controls subunit assembly to avoid mixing and matching ligand-selective subunits at will. I would tone this down - it's still speculative at this stage. We don't even know yet how this works for OR-Orco, where we do have structures. There is not yet an OR-Orco Cryo-EM structure, so we do not know what the subunit stoichiometry is.

We are not sure what the reviewer’s concern is. While direct biochemical or biophysical evidence is currently lacking, there is strong genetic evidence for heteromeric composition of Gr complexes, both from studies of bitter and sweet receptors/neurons (see response above). It is likely that intrinsic properties facilitate assembly of certain Grs within a taste receptor complex. We have refrained from any speculation about stoichiometry, though given the relatedness of Grs and Ors, it would not be far-fetched to propose that taste receptor complexes are also tetrameric in nature, which was recently proposed for a homomeric channel of the bombyx mori homolog of Gr43a, BmGr9 (Morinaga et al., 2022).

1. Line 305: the work of Emily Troemel and Cori Bargmann PMID: 9346234 should be cited in the Discussion. Theirs was the first experiment to show that valence was a feature of the neuron and not the receptor(s) it expresses.

We have now cited this work in the discussion to acknowledge this important discovery.

1. Figure 1 - the clarity of the organization of the figure could be improved for non-experts. For instance, can the key for the abbreviations be written out at the right of Figure 1A? Second, it is confusing to talk about DOG/TOG neurons "projecting" to the DO/TO - I think the authors mean dendritic innervation, not axons projecting. Maybe having a diagram that cartoons a closeup of the DOG/TOG neurons and how they innervate the cuticular structures would make this clearer. I struggled to go from the pretty staining at the left of B and C to the schematics at the right that colored in which neurons express which receptors.

We appreciate these comments regarding clarity and have amended Figure 1 and made necessary changes in the text and the Figure legend.

1. Figure 3 would benefit from a summary cartoon relating back to the cartoons in Figure 1 to summarize what neurons the authors think are necessary for bitter avoidance.

We very much appreciate this suggestion and have increased clarity by referring to the carton in Figures 1 and 2.

1. Figure 4B - the lowercase letters indicating Gr28 subunits that are being expressed under UAS control (bottom row of table "UAS-Gr28") are easily confused for the lowercase letters a, b used throughout to signify significant differences. I recommend that the authors write out the gene names in this figure to clarify the genes in the rescue experiment.

We changed the text in the Figure accordingly.

1. For non-experts it would be helpful to have a map of the Gr28 gene locus so that people understand the arrangement of the genes and how the Gal4 driver lines map onto the locus.

We have now included such a map in Figure 1B.

**Reviewer #2 (Recommendations For The Authors):**
1. In the title and multiple times in the text (e.g. lines 121-122), the authors make the claim that different Gr28 genes mediate opposing behaviors. At first, I was not convinced of this claim, but I now believe it may be warranted if integrating the present results with results from Mishra et al., 2018. In the present study, the authors show that different neurons drive opposing behaviors, but they did not show that the genes themselves mediate opposing behaviors. They show evidence for the role of Gr28bc and Gr28ba in aversion, but not the role of Gr28a in attraction. I was thinking that there could be other receptors in Gr28a-expressing neurons that mediate attraction. However, Mishra et al. showed that mutation of all Gr28 genes abolishes preference for RNA/ribose as well as detection of these compounds by Gr28a+ neurons of the terminal organ, an impairment that could be rescued by expressing Gr28a (although Gr28b genes seem to have similar functions), and the present study shows that the other Gr28 genes are not co-expressed with Gr28a in the terminal organ. Is this the line of reasoning that we must take to come to the conclusion in the title? If so, I don't believe it comes through clearly in the paper.

We appreciate this observation. We have modified language in the abstract and the introduction to reflect previous reports of Gr28a as an RNA/ribose receptor (Mishra et al., 2018) and its conversation across dipteran insects (Fujii et al., 2023) where we showed that appetitive behavior for RNA can be mediated via the mosquito homologs in transgenic *Drosophila* larvae. The reviewer is correct in that there are other appetitive neurons, namely those expressing Gr43a, which defines a set distinct from and non-overlapping with Gr28a neurons (Mishra 2018). This additional information is included in the Figure 1, summarizing expression of the Gr28 genes, Gr66a and Gr43a.

1. The Figure 6 schematic does not show Gr66a+ Gr28- cells as being connected to avoidance behavior. This seems misleading because it seems likely that these cells do promote avoidance (based on known functions of other Gr66a cells). Also, it is not clear what the red dashed line represents.

The Gr66a neurons are indeed also avoidance mediating, but it is not clear which subgroup of these neurons is necessary. Our analysis in Figure 2 using Gr28b.c driving Kir2.1 suggests that a small subset of Gr66a neurons is sufficient to mediate avoidance. It is, however, possible that other subsets not including Gr28b.c can also mediate avoidance. The figure has been modified accordingly, as has the model in Figure 7.

1. I would suggest including the description of Figures 7-8 in the Results instead of the Discussion. In Figure 8, it would be helpful to superimpose labels for the transmembrane domains and extracellular/intracellular sides to better interpret the models.

The modeling was removed from the manuscript (see response above to reviewer 1).

1. The finding that Gr66a mutants show increased denatonium and quinine avoidance (Figure 4 - figure supplement 1) seems like a non sequitur, as it does not relate to the analysis of Gr28 genes. I support the inclusion of these interesting results, but perhaps it could be stated why this experiment was conducted (e.g. as a positive control).

We have reworded this section to make clear why Gr66a mutants were tested (possibly being part of a denatonium receptor complex).

1. An introduction to the nomenclature and gene structure for the Gr28 genes would be helpful. It's not clear how they're all related, e.g. that the Gr28b genes share some exons whereas Gr28a is separate. The Results section alludes to "the high level of similarity between these receptors", and some sort of reference or quantification for this statement would be useful. I also think naming the Gr28b genes with a period (e.g. "Gr28b.c") may be more consistent with the literature.

We have added the structure of the Gr28 genes in the Figure 1B, which was also a suggestion by reviewer 1, and we have amended the naming of the genes.

1. Lines 79-80 state "some GRNs express members of both families", but no citation is provided.

As this sentence was deleted, based on a comment by reviewer 1, this point becomes mute.

1. There are several typos or grammatical mistakes that the authors may wish to correct (e.g. lines 73, 75, 91, 232, 334, 780, 788).

We appreciate the reviewer pointing these errors out to us. The mistakes were corrected.

**Reviewer #3 (Recommendations For The Authors):**
Silencing experiments suggest a role for Gr28bc in the avoidance of quinine (Figure 3), while imaging experiments do not support this role (Figure 5G). An explanation is needed to reconcile these findings.

The imaging experiments do support a role for Gr28b proteins in quinine detection in the specific TOG GRN used for all live imaging (Figure 5). This GRN in DGr28 larvae has a significantly lower Ca2+ responses to quinine compared to controls. However, the Ca2+ response could not be rescued to wild type levels by supplementing single Gr28b subunits, suggesting multiple Gr28b proteins are present in a quinine specific receptor complex in this GRN. Also note that Ca2+ responses of DGr28 larvae to quinine is not completely abolished, suggesting some redundancy, possible via Gr33a (Apostolopoulou et al., 2014), also supported by DGr28 larvae, which have still a robust avoidance to quinine. We are confident we have been clearer in arguing this point, both the result and especially the discussion section.

Silencing experiments specifically targeted neurons expressing Gr28bc and Gr28be (Figure 3). It is important to note why other neurons expressing different members of the Gr28 family were not included in this analysis.Inconsistency is observed in the use of different reagents across the experiments. Specifically, all six Gal4 lines were utilized in the Chemical Activation experiments, while only two lines were employed in the silencing experiments.

The silencing experiments asked the specific questions as to what neurons are necessary for avoidance of bitter chemicals. Gr28a-GAL4 and Gr28b.a-GAL4 neurons were omitted because the former mediate feeding preference and not avoidance, and the latter is expressed in the same neurons as Gr28b.e (Figure 1). The remaining two Gr28b genes, Gr28b.b-GAL4 and Gr28b.d-GAL4 are not expressed in the larval taste system (Mishra et al., 2018) as we stated in the introduction/result section, and they were therefore not included in the chemogenetic or Kir2.1 inactivation experiments. We included these genes in rescue experiments, simply to test whether or not they can restore function for sensing denatonium.

As for the chemogenetic activation experiments: two of the GAL4 lines are controls (Gr66a-GAL4 and Gr43GAL4), that were needed to show what can be expected from these experiments.

The authors did not acknowledge that neurons expressing members of the GR28 family also express other Gr family members, which could potentially contribute to the detection and behavioral responses to the tested bitter compounds.

We believe we did, but we have made that much more explicit in the revised manuscript.

Gal4 lines from various studies exhibit varying expression patterns, highlighting the necessity for improved reagents. These findings also suggest the importance of employing different Gal4 lines for each receptor to validate the results of the current study.

See response at the beginning of our rebuttal.

Activating or silencing neurons pertains to the function of the neurons rather than the receptors.

We agree and nothing in the manuscript states otherwise.